# Potassium ion homeostasis modulates mitochondrial function

Adam James Waite[1], Beiduo Rao[1], Elizabeth Schinski[1], Nathaniel H. Thayer[1], Manuel Hotz[1], Austin E. Y. T. Lefebvre[1], Celeste Sandoval[1], and Daniel E. Gottschling[1]

**Age-associated decline in mitochondrial membrane potential (MMP) is a ubiquitous aspect of eukaryotic organisms and is associated with many aging-related diseases. However, it is not clear whether this decline is a cause or consequence of aging, and therefore whether interventions to reduce MMP decline are a viable strategy to promote healthier aging and longer lifespans. We developed a screening platform in *Saccharomyces cerevisiae* to identify mutations that slowed or abrogated the age-associated decline in MMP. Characterization of the longest-lived mutant revealed that reduced internal potassium increased MMP and extended lifespan. Distinct interventions improved cellular MMP and lifespan: deleting a potassium transporter; altering the balance between kinases and phosphatases that control potassium transporter activity; and reducing available potassium in the environment. Similarly, in isolated mitochondria, reducing the concentration of potassium was sufficient to increase MMP. These data indicate that the most abundant monovalent cation in eukaryotic cells plays a critical role in tuning mitochondrial function, consequently impacting lifespan.**

## Introduction

The decline of mitochondrial function with age is observed across eukaryotes (Somasundaram et al., 2024; Chistiakov et al., 2014; Lima et al., 2022). This decline in function has been associated with cellular-level disorders such as reduced membrane potential, calcium cycling, and ATP production, increased production of reactive oxygen species, and loss of proteostasis (Sun et al., 2016; Haynes and Hekimi 2022; Pi et al., 2007). In metazoa, aging-related loss of mitochondrial function has been found in many tissues (Somasundaram et al., 2024) and is associated with general disorders such as decreased metabolic and muscle function, increased inflammation, and stem cell senescence (Sun et al., 2016; Grevendonk et al., 2021), as well as specific aging-related diseases of the lung (Cloonan et al., 2020), heart (Chistiakov et al., 2014), and brain (Coskun et al., 2012).

The complexity of mitochondrial biology and its connection to other subcellular systems (Gottschling and Nyström 2017) means that many biological processes currently not associated with mitochondrial health or function could nevertheless affect mitochondrial function within the context of the cell as a whole. For instance, a screen for genes that delayed the age-associated fragmentation of mitochondria found that this dysfunction was preceded by a loss of acidity in the lysosome-like vacuole in the budding yeast *Saccharomyces cerevisiae* (Hughes and Gottschling 2012). Mitochondria also engage in interorganelle communication through a wide variety of contact sites (Eisenberg-Bord et al., 2016). Loss of function in any of these other subcellular systems could directly affect mitochondrial function through a reduction or absence of these points of communication (Casler et al., 2025).

Of all the ways mitochondrial function can decline, the central role of the mitochondrial membrane potential (MMP) makes it a particularly good candidate for monitoring mitochondrial health (Nicholls 2004). In healthy cells, the MMP is primarily generated by the electron transport chain (ETC), but it is also influenced by any changes within the cell that alter the charge balance across the mitochondrial inner membrane (Zorova et al., 2018). Thus, the MMP integrates the state of cellular ion homeostasis, including the amounts and relative concentrations of charged molecules and ions in the cytosol relative to the mitochondrial matrix. These cellular properties are in turn determined by the amount and activity of channels and transporters (Palmieri and Monné 2016; Szabo and Szewczyk 2023).

The MMP is used to perform essential cellular functions. While generation of ATP via oxidative phosphorylation is probably the most well-known use of MMP, the mitochondria itself could not function without the MMP that it generates, as nearly all of the ~1,000 proteins necessary for mitochondrial function are encoded in the nuclear genome, translated in the cytosol, and imported into the mitochondria in an MMP-dependent manner (Kutik et al., 2007; Sickmann et al., 2003).

..................................................................................................................................

[1]Calico Life Sciences LLC, South San Francisco, CA, USA.

Correspondence to Daniel E. Gottschling: dang@calicolabs.com.

The molecular precursors necessary to generate ATP, such as pyruvate, ADP, and inorganic phosphate, all rely on the MMP-dependent transport to enter the mitochondria (Palmieri and Monné 2016), while the products of mitochondrial metabolism, such as ATP, nonessential amino acids, and other metabolites, rely on MMP-dependent export to be utilized by other parts of the cell (Palmieri and Monné 2016). Finally, apart from its role as a readout of overall mitochondrial health, the MMP appears to be a relevant target for intervention. Recent work in *Caenorhabditis elegans* demonstrated that artificially maintaining a high MMP using optogenetic tools was sufficient to improve mitochondrial function during stress and increase worm lifespan (Berry et al., 2020; Berry et al., 2023).

In an attempt to discover new pathways contributing to mitochondrial function, we developed a genome-wide screening platform based on MMP. With the information from this screen, we sought to find novel ways of intervening to delay or prevent the age-associated decline in MMP.

## Results

### MMP of young cells is correlated with their lifespan

To investigate the relationship between MMP and lifespan, we genomically integrated a fluorescent protein–based, two-component reporter system (Fig. 1, A and B; Video 1) into *S. cerevisiae* (Materials and methods). This system contained an MMP-sensitive ($MMP_s$) component and an MMP-insensitive ($MMP_i$) component to allow longitudinal, ratiometric measurement of MMP in live cells, and circumvents many limitations of dye-based MMP reporters such as tetramethylrhodamine methyl ester (TMRM). The dyes cannot be fixed for fluorescence-activated cell sorting (FACS) analysis and cannot be left in live cells for extended periods of time, and their signal does not take into account changes in mitochondrial mass or the amount of dye taken up by cells, which is determined by the plasma membrane potential (Scaduto and Grotyohann 1999; Brand and Nicholls 2011).

The expression of the $MMP_s$ component is driven by the promoter of *TPI1* and contains the leader sequence of *COX4* ("preCOX4") fused to the fluorescent protein mNeonGreen and the degron *SL17* (Gilon et al., 1998). Import of preCOX4-tagged fluorescent proteins is well characterized and specifically depends on MMP (Veatch et al., 2009; Fehrmann et al., 2013; Garipler et al., 2014; Vowinckel et al., 2015), including throughout replicative lifespan (Fehrmann et al., 2013). The *SL17* degron is a well-characterized signal sequence that confers a half-life of minutes in the cytosol (Gilon et al., 1998), but is completely protected from degradation in the mitochondrial matrix (Shlevin et al., 2007; Papić et al., 2013), thus preventing accumulation of the mNeonGreen signal in the absence of MMP (Garipler et al., 2014; Vowinckel et al., 2015).

The $MMP_i$ component is a C-terminal mCherry fluorescent tag of the mitochondrial inner membrane protein Tim50p, which is part of the Translocase of the Inner Mitochondrial membrane complex (TIM23) (Kutik et al., 2007). Tim50p is essential (Yamamoto et al., 2002), accurately represents mitochondria in aged cells lacking MMP ([Hughes and Gottschling 2012], Fig. S1, B and E), and, unlike mitochondrial outer membrane proteins, is not found in aging-related, vacuolar-associated, mitochondrial-derived compartments (Hughes et al., 2016). These attributes make Tim50p a good $MMP_i$ reporter of mitochondrial mass. We used the ratio of the mNeonGreen signal to the mCherry signal ("MMP ratio") as a readout of the cell's MMP-dependent import capacity normalized by its mitochondrial mass.

We used the Yeast Lifespan Machine (YLM [Thayer et al., 2022, *Preprint*]) with fluorescence images taken every 2 h to track the MMP ratio of ~12,000 cells as they aged (Fig. 1, C and D; Fig. S1; Materials and methods). While there was significant heterogeneity in individual cell trajectories, we found that, similar to Hughes and Gottschling (2012), but in contrast to Fehrmann et al. (2013), the MMP ratio of every cell declined with age. This decline in MMP ratio was driven by a decrease in the $MMP_s$ signal (Fig. S1), and occurred regardless of whether it was measured by time (Fig. 1 C, Fig. S1, first column), number of divisions (Fig. 1 D, Fig. S1, second column), number of divisions before death (Fig. S1, third column), or fraction of divisions remaining (Fig. S1, fourth column). Using an orthogonal, microscopy-based imaging assay, we found that relative to a new mother cell, TMRM declined more rapidly with age than $MMP_s$ (Fig. S2 B). Thus, the $MMP_s$ reporter is somewhat delayed in responding to reduction of MMP with age. Interestingly, there was a significant correlation between a cell's MMP ratio averaged over its first five divisions and the number of divisions it completed before death ($r = 0.55$, $p \sim 0$; Fig. 1 E), suggesting that higher initial MMP is associated with a longer lifespan.

### A screen for suppressors of the age-associated decline of MMP

In order to discover processes that could delay or prevent the age-associated reduction in MMP, we set up a screening platform to identify loss-of-function mutants that altered MMP in aged cells (Fig. S3 A). Specifically, we introduced the reporter system (Fig. 1 A) into the yeast knockout collection (Winzeler et al., 1999), in which each of the ~4,700 nonessential genes has been deleted and tagged with a unique barcode (Materials and methods). This deletion library was pooled and aged using Ministat Aging Devices (MADs) (Hendrickson et al., 2018) for 24 h (mean age 15.1 divisions; Fig. S3 C), a time at which wild-type (WT) cells have a dramatically reduced MMP as evaluated by either TMRM or the MMP reporter ratio (Fig. 1 C, Fig. S2 B). We then used FACS to sort mother cells from the bottom, middle, or top thirds of the distributions of the MMP ratio (calculated from the green/red signals) (Fig. S3 B; Materials and methods).

We focused our attention on the mutants enriched in the high MMP ratio group relative to the low MMP ratio group. Deletions of the top 35 mutant genes from the screen were remade in a prototrophic strain background that contained the MMP reporter system (Table S5). Using the YLM, we tracked the MMP of mother cells in each of these strains. Statistical modeling of the trajectories (Materials and methods) revealed that 26 (74%) of the deleted genes identified in the screen had a higher MMP ratio than WT after 24 h in the microfluidics device (at a significance threshold of $P < 0.05$), indicating good concordance between the screen and microfluidics approaches. In young cells, TMRM

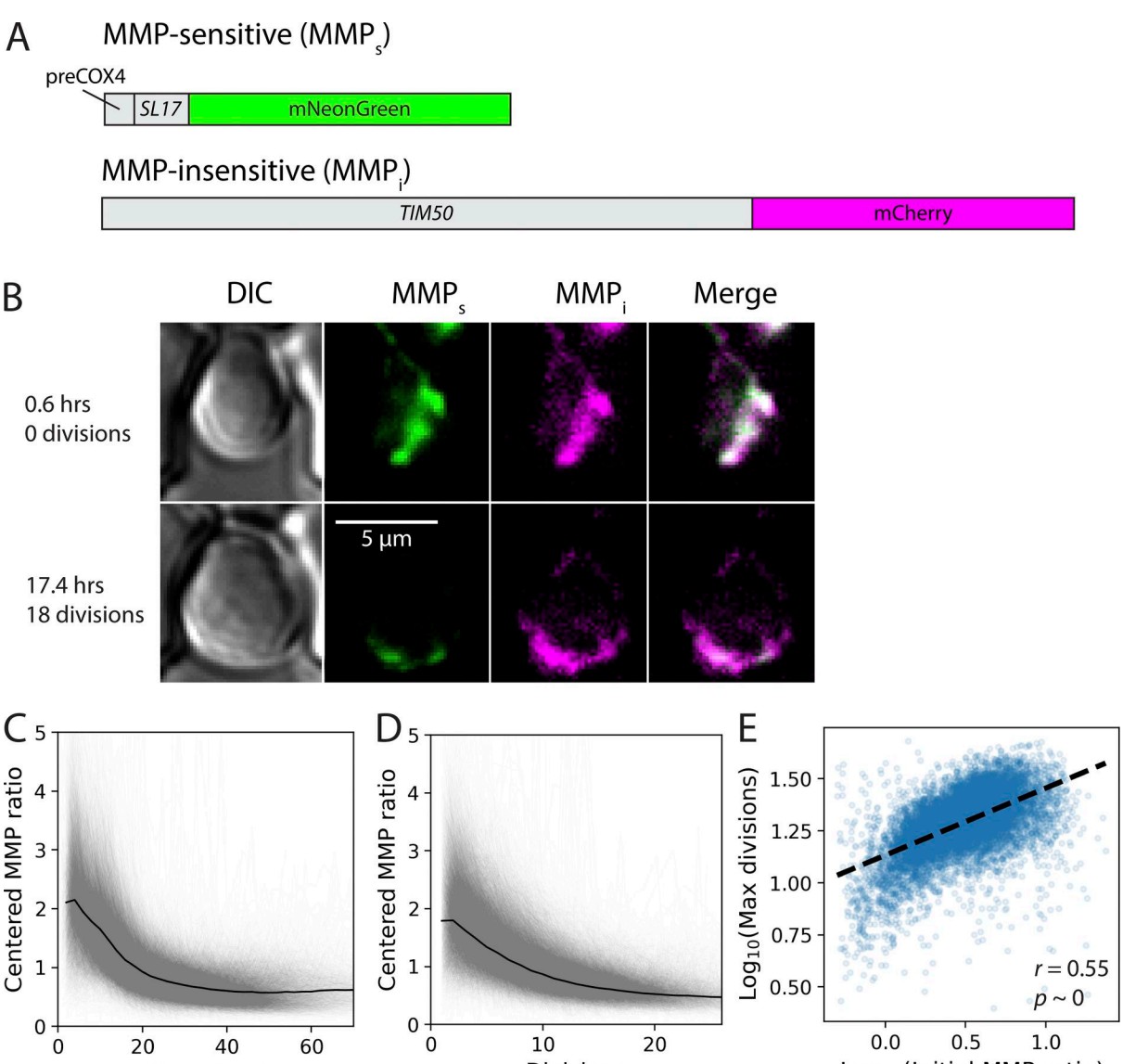

Figure 1. **The MMP ratio is correlated with lifespan and declines with age in WT cells. (A)** Schematic of the two-component construct. **(B)** Representative composite fluorescent images taken from the YLM showing the expression of the reporter construct in the same mother cell when young (top) and old (bottom). Overlap of the MMP$_s$ (green) and MMP$_i$ (magenta) signals appears white. The full set of fluorescence images associated with this cell can be seen in Video 1. **(C and D)** Trajectories of the MMP ratio (MMP$_s$/MMP$_i$) as cells age. Each gray line represents the centered (i.e., mean-normalized) MMP ratio trajectory of a single cell. Trajectories are from cells that were observed to die in the device (n = 11,856 from 17 experiments). Uncentered trajectories can be found in Fig. S1. Black line indicates the mean trajectory. **(C)** Centered MMP ratio as a function of time. **(D)** Centered MMP ratio as a function of number of divisions. **(E)** Number of divisions achieved before cell death as a function of the MMP ratio when young (0–5 divisions). Data are from the same cells analyzed in C and D. r is the Pearson correlation coefficient, and p is the P-value of the estimate of r.

accumulation was strongly correlated with the reporter MMP$_s$ signal that was normalized to total mitochondrial mass (Pearson's r = 0.79, P = 1.2 × 10$^{-7}$; Fig. S3 F), thus providing independent evidence for the success of the MMP reporter and its utilization in this screen.

**Deleting *SIS2* promotes mitochondrial function and increases lifespan**
Of the 26 deletion mutants confirmed to have a higher MMP ratio than WT after 24 h, 21 had significantly longer lifespans than WT, but *sis2*Δ was by far the longest lived (Table S2).

Despite several observations that deletion of *SIS2* extends lifespan (McCormick et al., 2015; Ölmez et al., 2023; Smith et al., 2008), the mechanism by which this lifespan extension occurs remains unclear, and its connection to mitochondrial function has not been reported. The lifespan extension of the *sis2*Δ mutant was additive with one of the most well-characterized lifespan-extending deletions, *fob1*Δ, suggesting that the mechanism is not related to the extrachromosomal ribosomal DNA circle–mediated mechanism of lifespan determination (Sinclair and Guarente 1997) (Fig. 2, A and B). The MMP ratio trajectory of the *sis2*Δ mutant declined only slightly throughout the lifespan

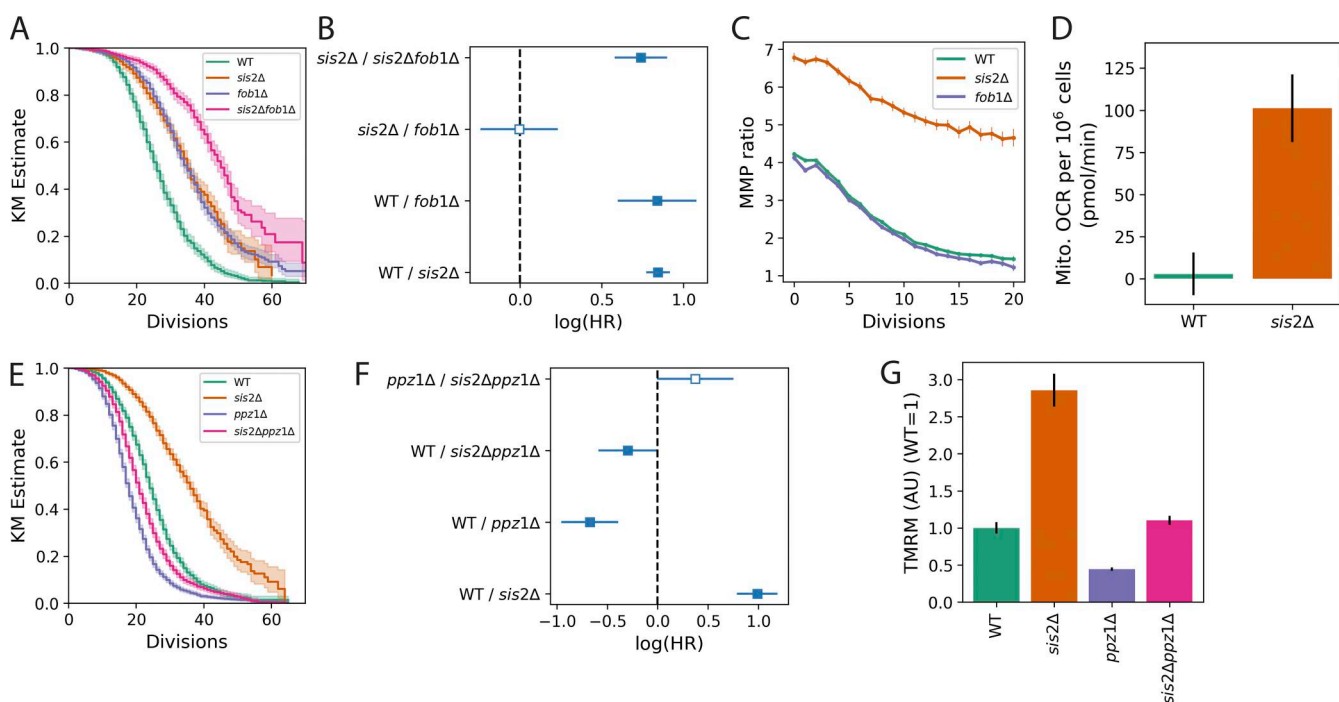

Figure 2. **Increased lifespan and MMP of the *sis2*Δ mutant are independent of *fob1*Δ and are blocked in a *ppz1*Δ background. (A)** Kaplan–Meier lifespan estimates showing the interaction between *sis2*Δ and *fob1*Δ. Data are from two experiments. *n* = 1,131 for WT, *n* = 1,123 for *sis2*Δ, *n* = 1,157 for *fob1*Δ, and *n* = 1,139 for *sis2*Δ*fob1*Δ. Shaded area is the 95% confidence interval. **(B)** Logarithm of the hazard ratio ("log[HR]") calculated using CPH modeling (see Materials and methods). The filled square indicates the log(HR) was significantly different from zero (at an adjusted P-value of 0.05); empty square indicates the comparison was not significantly different from zero. Error bars show 95% confidence intervals. **(C)** MMP ratio trajectories. WT data are averaged over 17 experiments (*n* = 15,610); *sis2*Δ data are averaged over four experiments (*n* = 5,111); *fob1*Δ data are averaged over eight experiments (*n* = 6,144). Error bars show two times the SEM. **(D)** Mitochondrial OCR as measured using Seahorse (Materials and methods). Data are a representative experiment of three independent experiments. *n* = 4 replicates per genotype per experiment. Error bars show 2 × SEM. **(E)** Kaplan–Meier lifespan estimates showing the interaction between *sis2*Δ and *ppz1*Δ. Data are from four experiments. *n* = 2,356 for WT, *n* = 2,237 for *sis2*Δ, *n* = 3,015 for *ppz1*Δ, *n* = 2,282 for *sis2*Δ*ppz1*Δ. **(F)** log(HR) from a CPH model of **E**. **(G)** MMP as measured by TMRM for the strains in **E**, normalized to the mean WT value. Data are from two experiments. In each experiment, three replicate measurements of 10,000 cells were made for each strain. Error bars show 95% confidence intervals. CPH, Cox proportional hazard; SEM, standard error of the mean.

of the cell and leveled off at a value higher than WT (Fig. 2 C, orange), while the *fob1*Δ mutant had no effect on the MMP ratio (Fig. 2 C, purple). The mitochondrial oxygen consumption rate (OCR) was also elevated in *sis2*Δ (Fig. 2 D), indicating that increased MMP in this mutant was associated with more active mitochondria (Schmidt et al., 2021; Nicholls 2004).

**sis2Δ aging phenotypes appear to act through Ppz1p**

Sis2p has been implicated in performing two distinct functions. It is a subunit of the heterotrimeric phosphopantothenoylcysteine decarboxylase complex involved in the biosynthesis of coenzyme A (Ruiz et al., 2009), and is an inhibitor of the serine/threonine phosphatase Ppz1p (de Nadal et al. 1998). We investigated the role of Sis2p in coenzyme A biosynthesis, as this has been proposed as a mechanism for the observed lifespan extension (Ölmez et al., 2023). When *SIS2* was deleted, targeted metabolomics showed no change in the amount of coenzyme A, the coenzyme A precursor pantothenate, or the coenzyme A product acetyl-CoA (Fig. S4, A and B; Table S3). This was consistent with a recent report indicating no change in these metabolites in the *sis2*Δ mutant (Ölmez et al., 2023).

On the other hand, we did find evidence that the observed MMP and lifespan phenotypes of *sis2*Δ were related to a loss of

Ppz1p inhibition. The overexpression of *PPZ1* results in hyperpolarization of the plasma membrane due to reduced potassium uptake (Ariño et al., 2010; Navarrete et al., 2010; Kahm et al., 2012; Herrera et al., 2014). This causes sensitivity to toxic cations such as lithium. Consistent with this idea, *sis2*Δ had a growth defect on 300 mM LiCl, though it was not as severe as a deletion of the gene encoding the high-affinity potassium transporter, *TRK1* (Fig. S4 C). In addition, we confirmed previous observations (Ölmez et al., 2023) that the *ppz1*Δ mutant had a shorter lifespan than WT and prevented the lifespan extension of *sis2*Δ (Fig. 2, E and F). Deleting *PPZ1* also caused a reduction in young cell MMP compared with WT and prevented the MMP increase of *sis2*Δ, as measured by TMRM (Fig. 2 G). Taken together, these data suggested that one of the ways Sis2p was impacting MMP and lifespan was via its regulation of Ppz1p. Specifically, we hypothesized that deleting *SIS2* resulted in increased activity of Ppz1p.

**Deletion of Npr/Hal-family kinases increases MMP and extends lifespan, suggesting a common target with Ppz1p**

One of the better characterized functions of Ppz1p is the destabilization of nutrient and ion transporters at the plasma membrane (Ariño et al., 2010; Lee et al., 2019). The Npr/Hal-family

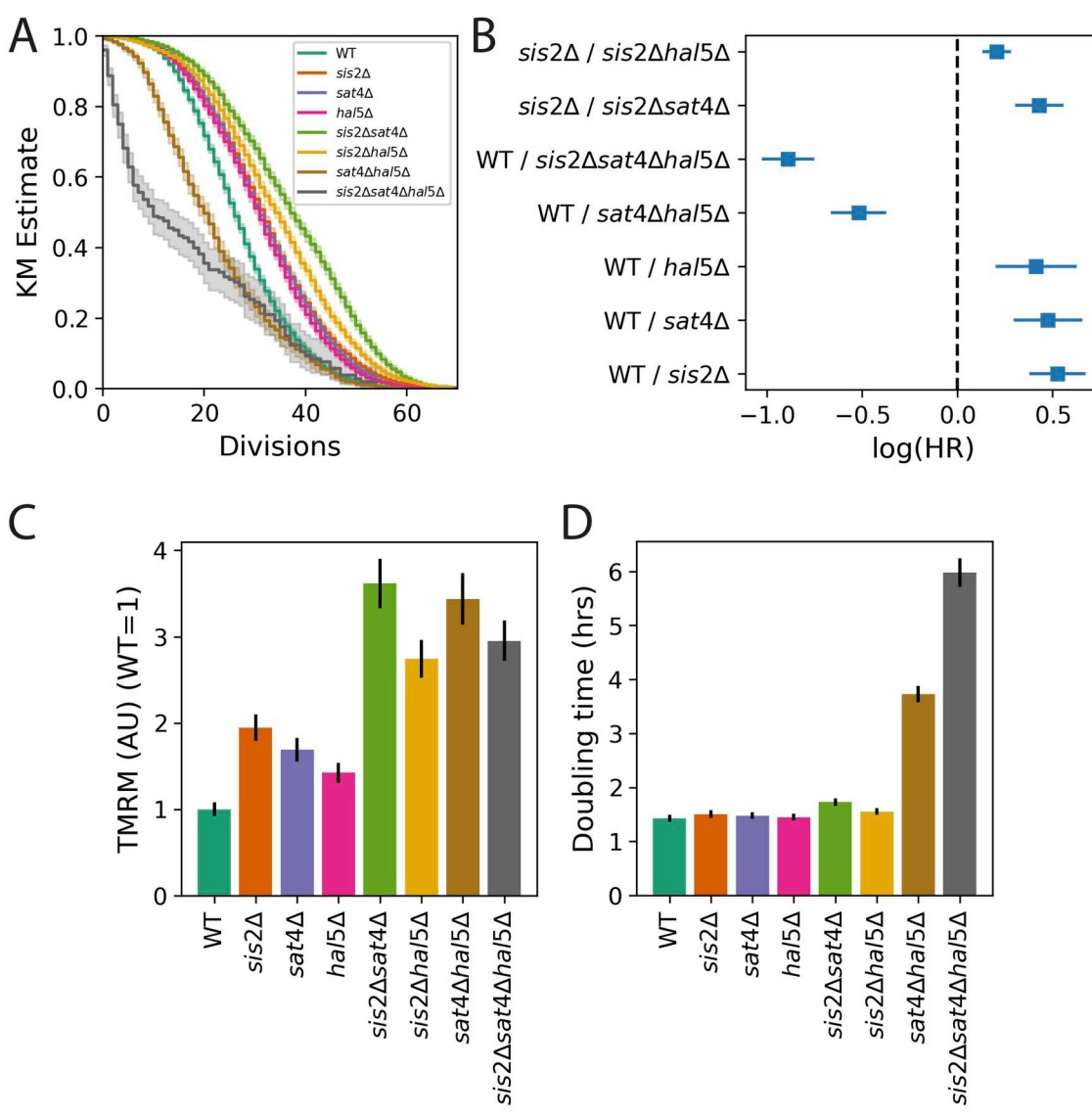

Figure 3. **Lifespan and MMP phenotypes of sat4Δ and hal5Δ mutants are additive with sis2Δ.** **(A)** Kaplan–Meier lifespan estimates. n = 6,500 for WT, n = 4,806 for sis2Δ, n = 6,874 for sat4Δ, n = 5,542 for hal5Δ, n = 4,563 for sis2Δsat4Δ, n = 4,874 for sis2Δhal5Δ, n = 2,348 for sat4Δhal5Δ, n = 535 for sis2Δsat4Δhal5Δ. Data are from three experiments, but technical issues reduced the representation of the sat4Δhal5Δ and sis2Δsat4Δhal5Δ strains to one unique experiment each. **(B)** log(HR) from a CPH model of A. See legend of Fig. 2 for details. **(C)** MMP for strains in A, as measured using TMRM. Values are normalized to the mean WT value. Data are from two experiments, 20–28 replicates per genotype and 10,000 cells analyzed per replicate. **(D)** Culture doubling times, as measured after 24 h of exponential growth before lifespans (A) and MMP (C), were measured. Data are from three experiments; 7–9 replicates per genotype. In all panels, error bars show 95% confidence intervals.

kinases Sat4p (aka Hal4p) and Hal5p (Antunes and Sá-Correia 2022) phosphorylate and stabilize many of the same targets that Ppz1p dephosphorylates and destabilizes (Mulet et al., 1999; Pérez-Valle et al., 2007; Tumolo et al., 2020). Thus, if the increased phosphatase activity of Ppz1p was required for the increased MMP and lifespan of a sis2Δ mutant, then removing one or more of these antagonistic kinases should also increase MMP and lifespan. (We did not obtain information about sat4Δ or hal5Δ mutants from our screen, as they were underrepresented in the screening pool of deletion mutants.) Consistent with this hypothesis, we found that deleting SAT4 or HAL5 increased MMP (P = 2.5 × 10⁻¹³ and 8 × 10⁻⁷, respectively) and extended lifespan (Fig. 3, A and B). For both gene deletions, increased MMP and

extended lifespan were additive with sis2Δ (Fig. 3 C, P = 1.2 × 10⁻¹³ for sis2Δ vs sis2Δ sat4Δ; p = 1 × 10⁻⁶ for sis2Δ vs sis2Δ hal5Δ). On the other hand, it has been shown that overexpressing Ppz1p can be toxic to the cell (Makanae et al., 2013; Casamayor and Ariño 2022), suggesting that unbalanced kinase/phosphatase activity favoring dephosphorylation by Ppz1p can be detrimental to the cell. Consistent with this, we and others (Pérez-Valle et al., 2010) found that the sat4Δ hal5Δ double mutant grew slowly (Fig. 3 D), and we showed that it had a short lifespan (Fig. 3, A and B). The results were even more extreme for the sis2Δ sat4Δ hal5Δ triple mutant (Fig. 3, A–D). Intriguingly, both the sat4Δ hal5Δ mutant and the sis2Δ sat4Δ hal5Δ mutant displayed an MMP higher than sat4Δ or hal5Δ alone (Fig. 3 C; all p < 2 × 10⁻¹³ for these

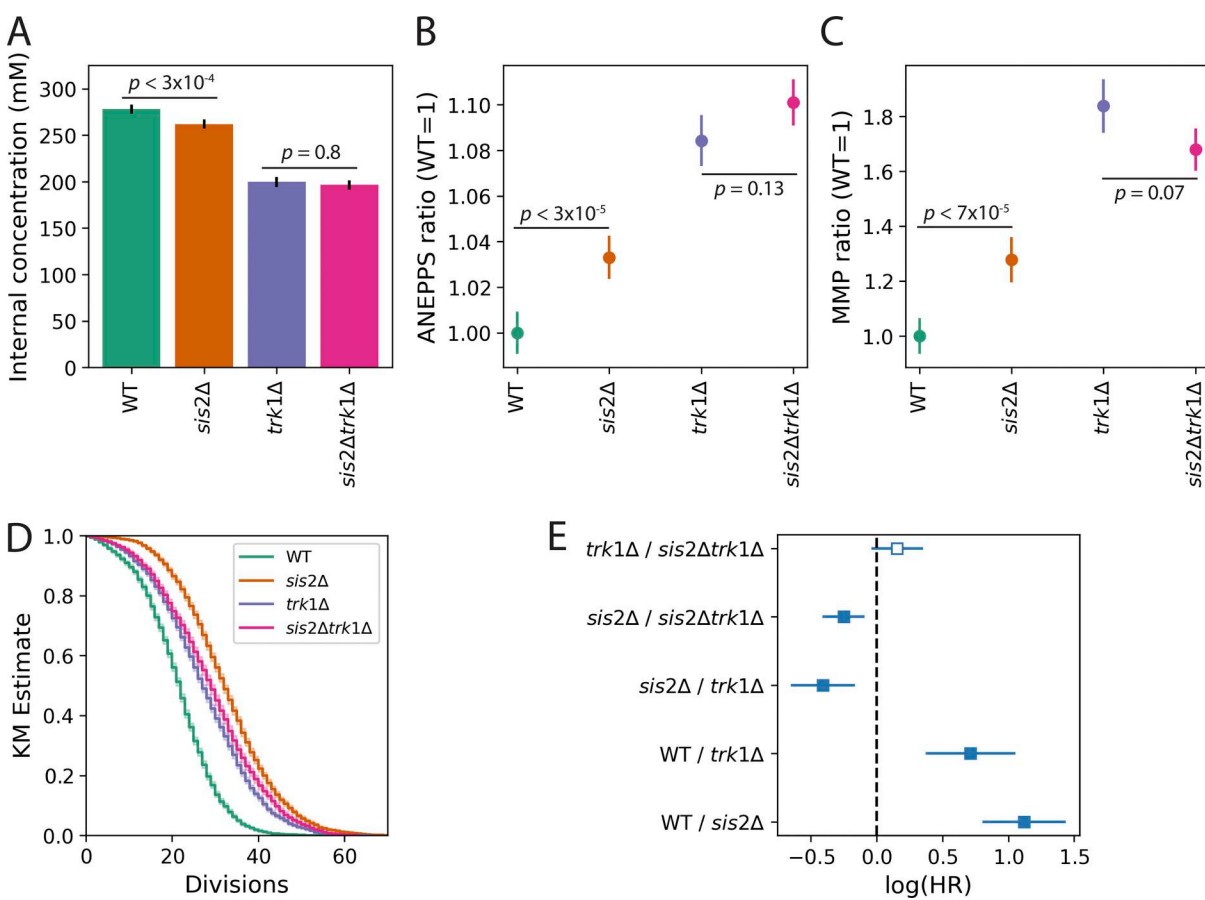

Figure 4. ***sis2Δ* mutant behaves like a hypomorphic *TRK1* allele.** **(A)** Internal potassium concentration. Data are from two independent experiments with three replicates per experiment per strain. **(B)** Plasma membrane potential, as measured by the ANEPPS ratio (see Materials and methods for details). Ratios are normalized to the WT value. Data are from five experiments with 3–4 technical replicates per genotype per experiment. **(C)** MMP ratio as measured by the genomic reporter (Fig. 1 A) and normalized to WT. Data are from two experiments with three technical replicates for each genotype. **(D)** Kaplan–Meier lifespan estimates. Data are from four (WT, *sis2Δ*, *trk1Δ*) or two (*sis2Δtrk1Δ*) experiments. *n* = 6,333 for WT, *n* = 5,871 for *sis2Δ*, *n* = 7,359 for *trk1Δ*, *n* = 4,632 for *sis2Δtrk1Δ*. **(E)** log(HR) comparisons of data from D. In all panels, error bars show 95% confidence intervals.

comparisons). Collectively, these results supported the hypothesis that the target protein or proteins responsible for modulating MMP and lifespan were shared between the kinases Sat4p and Hal5p and the phosphatase Ppz1p, and were thus likely to be nutrient or ion transporters.

## Increased MMP and extended lifespan are associated with reduced activity of the Trk1p transporter

One of the most well-characterized targets of Ppz1p that is also a target of Sat4p and Hal5p is the high-affinity potassium transporter, Trk1p (Ariño et al., 2010; Ariño et al., 2019; Yenush 2016). Since deletion of *TRK1* and potassium limitation have been shown to extend lifespan (Sasikumar et al., 2019), we speculated that *sis2Δ* could be acting as a *TRK1* hypomorphic allele. Consistent with this hypothesis, several *sis2Δ* phenotypes were intermediate between WT and *trk1Δ*, including lower internal potassium (Fig. 4 A), increased plasma membrane potential (Fig. 4 B and Fig. S4 C), and increased MMP (Fig. 4 C). In all cases, the phenotypes of the *sis2Δ trk1Δ* double mutant were not significantly different from the *trk1Δ* mutant (Fig. 4, A–C), suggesting that for these phenotypes, *SIS2* and *TRK1* are in the same

pathway. Deleting *SIS2* increased lifespan more than deleting *TRK1*, while the lifespan of the *sis2Δ trk1Δ* double mutant was equivalent to the *trk1Δ* mutant (Fig. 4, D and E). This is also consistent with *sis2Δ* acting as a *TRK1* hypomorphic allele, as it suggests that reducing Trk1p activity (by deleting *SIS2* only) is more beneficial to lifespan than eliminating Trk1p activity completely (by deleting *TRK1*).

## Potassium limitation increases MMP-dependent import capacity and extends lifespan

Since Ppz1p has other targets in addition to Trk1p (Velázquez et al., 2020), we tested whether potassium limitation was sufficient to increase MMP and extend lifespan. We grew WT and *sis2Δ* strains in media with different concentrations of potassium and measured their MMP ratios and lifespans (Fig. 5, dark colors and solid lines). The MMP ratio and lifespan of the WT and *sis2Δ* strains increased as external potassium was reduced (Fig. 5 A, dark colors and solid lines). At ∼0.5 mM external potassium, WT and *sis2Δ* had equivalent MMP ratios. Further reduction of external potassium to 0.1 mM caused a small decline in the MMP ratio (Fig. 5 A, dark colors and solid lines). Interestingly, both

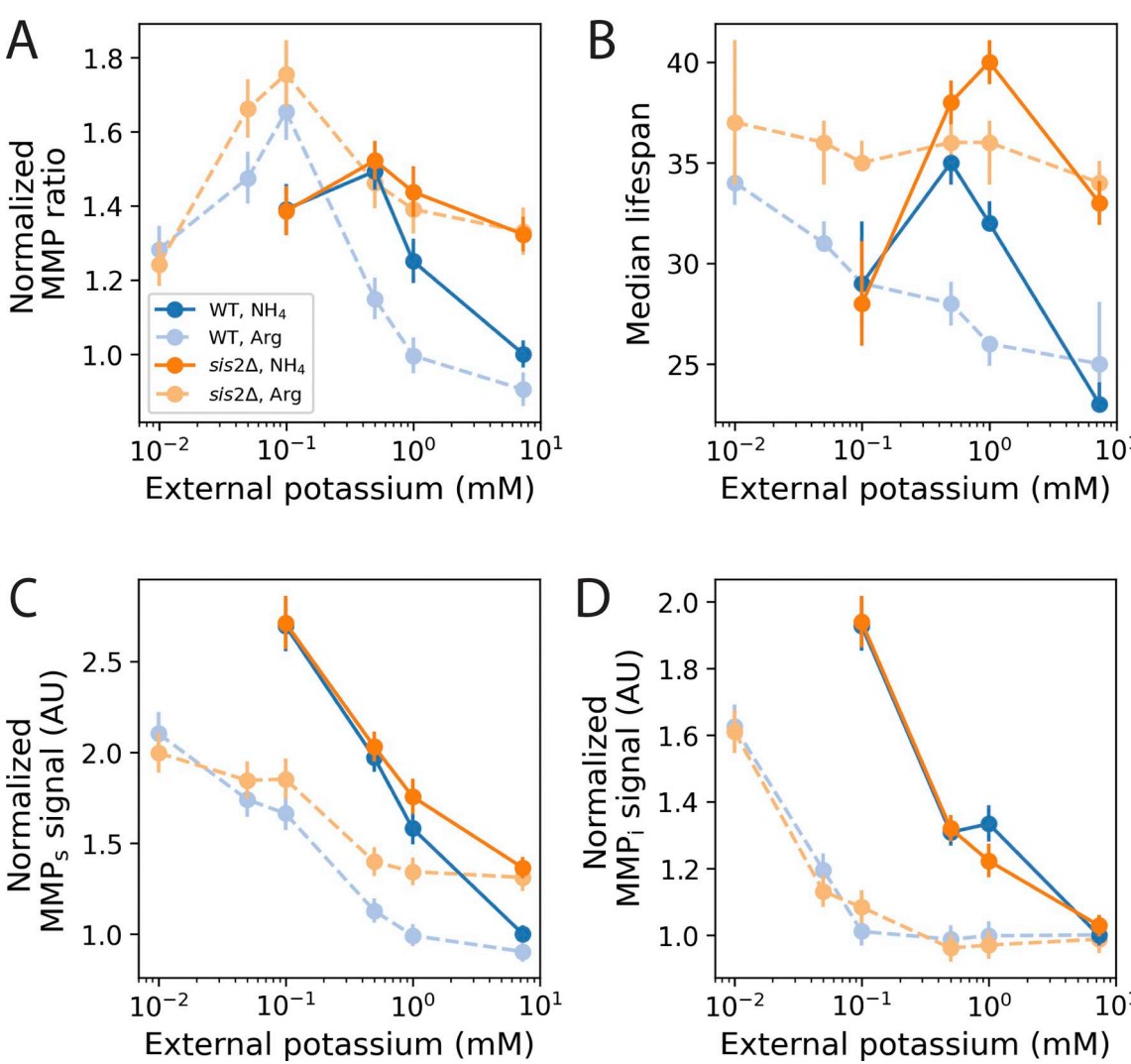

Figure 5. **Potassium limitation increases MMP and lifespan. (A)** MMP ratio as a function of external potassium for WT (blues) and *sis2Δ* (oranges) in either ammonium sulfate ("NH$_4$," dark symbols and dark, solid lines) or arginine ("Arg," light symbols and light, broken lines) as the nitrogen source. The MMP ratio is normalized to the value of WT in standard media (i.e., ammonium sulfate and 7.35 mM potassium). Results are from four experiments with ammonium sulfate and two experiments with arginine as the nitrogen source. $n \geq 6$ replicates for every strain, concentration, and nitrogen source combination, with 10,000 cells analyzed per replicate. **(B)** Median replicative lifespan as a function of external potassium, estimated from a CPH model (Materials and methods). Data are from three (NH$_4$) or two (Arg) experiments. $n \geq 1,400$ cells for every genotype, nitrogen, and potassium concentration combination. **(C and D)** MMP$_s$ (C) and MMP$_i$ (D) signals as a function of external potassium concentration, from the same experiments as A, normalized to WT in ammonium sulfate at 7.35 mM external potassium. All error bars show 95% confidence intervals.

MMP$_s$ and MMP$_i$ increased as external potassium was reduced (Fig. 5, C and D, dark colors and solid lines). The reduction of the MMP ratio at very low levels of potassium reflected a greater increase in MMP$_i$ relative to MMP$_s$.

Lifespan dramatically declined at 0.1 mM external potassium (Fig. 5 B, dark colors and solid lines). While this decrease in lifespan brought the *sis2Δ* strain below its lifespan in media with standard potassium, the lifespan of WT was still higher than its lifespan in standard potassium (Fig. 5 B, dark colors and solid lines).

While performing the initial experiments in ammonium sulfate, we observed a dramatic increase in culture doubling time as external potassium was reduced, and absolutely no growth below 0.1 mM external potassium (Fig. S4 D, dark colors

and solid lines). Slower culture growth in limiting potassium has been attributed to toxicity associated with excessive ammonium uptake (Hess et al., 2006; Rodríguez-Navarro and Ramos 1984) because ammonium, the main nitrogen source in standard media, competes with potassium for uptake by Trk1p (Conway and Duggan 1958). To remove this effect, we repeated potassium limitation experiments with arginine in place of ammonium as the primary nitrogen source (Fig. 5, light colors and broken lines). Below standard medium potassium concentrations, cultures doubled more quickly in arginine than in ammonium, and cultures in arginine even grew in 0.01 mM potassium (Fig. S4 D). However, culture doubling times increased even with arginine as a nitrogen source, demonstrating a growth defect in extreme potassium limitation (Fig. S4 D).

While arginine had different effects on both the MMP ratio (Fig. 5 A, light colors and broken lines) and lifespan (Fig. 5 B, light colors and broken lines) than ammonium, growth in limiting potassium was sufficient for the WT strain to phenocopy the high MMP ratio and long lifespan of a *sis2Δ* mutant grown at the standard potassium concentration regardless of the nitrogen source. As in ammonium, growth in arginine and limiting potassium resulted in a large increase in mitochondrial mass (as measured by MMP$_i$) to support increased MMP$_s$ import (Fig. 5, C and D).

**The increased MMP and lifespan extension in limiting potassium do not appear to act through vacuolar acidification**

The lifespan extension of potassium limitation has been attributed to increased acidification of the yeast lysosome-like vacuole (Sasikumar et al., 2019). We checked vacuolar acidification using a fluorescence protein–based, ratiometric, genomically integrated pH sensor specifically designed to report on vacuolar pH (Okreglak et al., 2023), and found that in fact, the vacuole generally became less acidic (more alkaline) when reducing external potassium below standard medium levels (Fig. S4). In our conditions, there was no difference between WT and *sis2Δ* vacuolar acidity at 1 mM external potassium (Fig. S4 E), despite large differences in the MMP ratio (Fig. 5 A) and lifespan (Fig. 5 B). Thus, we sought other explanations for the observed effects on MMP and lifespan.

**Internal potassium concentration is correlated with MMP$_s$ import capacity, which is correlated with lifespan**

The results so far demonstrated that changing external potassium concentration led to changes in MMP. Since in yeast (Kahm et al., 2012; Herrera et al., 2014; Masaryk and Sychrová 2022) and mammalian cells (Patrick 1978; Relman et al., 1961), internal and external concentrations of potassium are correlated, we hypothesized that the internal concentration of potassium might, directly or indirectly, determine MMP. We therefore measured the internal potassium of WT and *sis2Δ* strains grown in different potassium concentrations, in either ammonium or arginine. We confirmed that in both strains, lower external potassium resulted in lower internal potassium, regardless of nitrogen source (Fig. 6 A). Consistent with the idea that ammonium competes with potassium uptake, the decline in internal potassium was greater for cells grown in ammonium (Fig. 6 A). There was a general anticorrelation between MMP$_s$ and external potassium (Fig. 5 C; Pearson's $r = -0.67$, P = 0.001), but to a different degree depending on nitrogen environment and genotype (Fig. 5 C). In contrast, there was a very strong anticorrelation between MMP$_s$ and internal potassium, which held across nitrogen environments and genotypes ($r = -0.98$, $p < 4 \times 10^{-9}$; Fig. 6 B). Increasing external potassium to 200 mM did increase internal potassium in both WT and *sis2Δ* strains (Fig. S5 A), but this had a negligible effect on MMP$_s$ (Fig. S5 B) or lifespan (Fig. S5 C), suggesting some other processes, such as vacuolar sequestration (Herrera et al., 2013) or alternative feedback mechanisms, alter this relationship when external potassium is high. We also found that aged WT cells had a lower concentration of potassium than young cells (Fig. S5 E). Thus, for young

cells growing in standard to limiting amounts of potassium, internal potassium determined MMP$_s$ import capacity, regardless of genotype or nitrogen source.

In young WT cells grown in arginine as a nitrogen source, potassium-dependent increases in MMP correlated very well with lifespan (Fig 6 C). In *sis2Δ* cells, the lifespan was greater than WT at all MMP$_s$ levels, suggesting that Sis2p increases lifespan by additional pathways. To investigate why *sis2Δ* cells have longer lifespans than WT in all external potassium concentrations (Fig. 5 B), we reasoned that genes whose expression was significantly higher or lower than WT in 7.35 and 0.5 mM KCl would be good candidates for the observed MMP- and potassium-independent lifespan extension of *sis2Δ*. As observed previously (Ölmez et al., 2023), the expression of most ETC transcripts was elevated in *sis2Δ* compared with WT in 7.35 mM KCl (Fig. S5 F). However, in 0.5 mM KCl, the expression of ETC transcripts in WT cells increased, such that they were not significantly different between WT and *sis2Δ*, with the exception of COX6 (Fig. S5 G). Thus, the expression of ETC components was unlikely to account for the extended lifespan of *sis2Δ* in low potassium. Overall, the expression of 61 genes was differentially expressed between *sis2Δ* and WT in both environments (Table S4). Among these, only the set of 22 genes lower in both conditions had any significant functional enrichments, and they were all related to amino acid biogenesis (Fig. S5 H). Consistent with this, of the 23 metabolites found to be significantly lower in *sis2Δ* vs WT, 13 were amino acids or amino acid intermediates (Table S3). This altered metabolism may contribute to, or reflect processes involved in, its longer lifespan in all potassium concentrations.

**Potassium concentration determines MMP of isolated mitochondria by altering the ionic strength of the mitochondrial environment**

To determine whether potassium ion itself was directly responsible for altering MMP, we isolated mitochondria from cells and monitored TMRM in different conditions (Scaduto and Grotyohann 1999). We titrated the amount of KCl in the incubation buffer from 40 to 250 mM (Fig. 7 A), corresponding to the range of values observed *in vivo* (Fig. 6 B). Electron transport was stimulated by the addition of the metabolic substrates glutamate/malate, succinate, and glycerol-3-phosphate (Fraenkel 2011). MMP declined with increasing KCl in the buffer (Fig. 7 A, green), suggesting that the KCl concentration had a direct effect on MMP. This effect persisted after adding ADP to stimulate complex V, demonstrating that the isolated mitochondria were fully functional and capable of oxidative phosphorylation at all KCl concentrations (Fig. 7 A, orange) (Nicholls 2004). The addition of carbonyl cyanide-*p*-trifluoromethoxyphenylhydrazone (FCCP), which effectively makes the mitochondrial inner membrane permeable to protons (Nicholls 2013), reduced MMP to an equivalent level across all KCl concentrations (Fig. 7 A, purple), providing further evidence that we were monitoring MMP. These results were also seen for the *sis2Δ* mutant (Fig. S5 I). The effect of KCl on MMP was rapid and reversible: mitochondria preincubated in 250 mM KCl for 15 min and then diluted into a

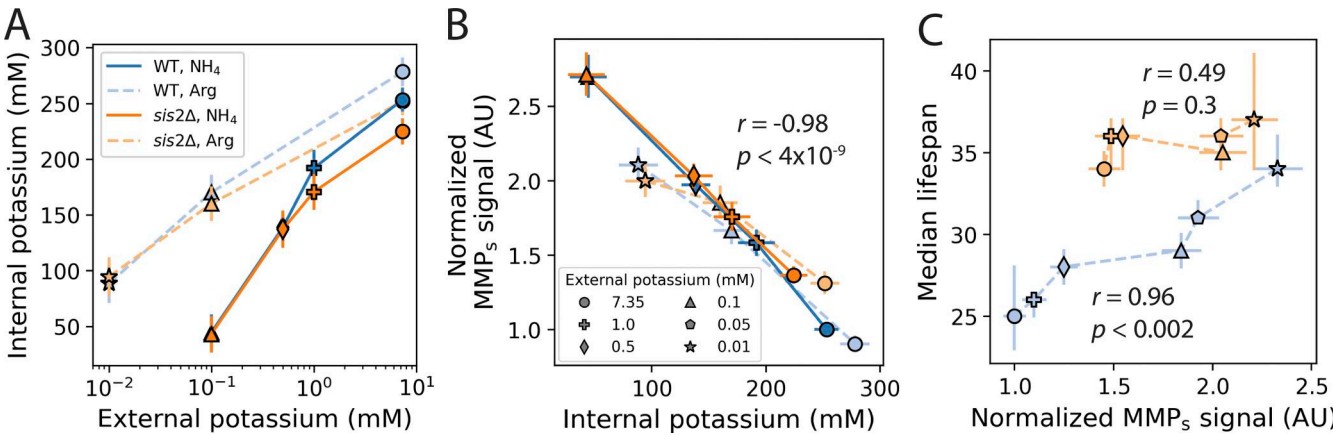

**Figure 6.** **Internal potassium determines MMP$_s$, and MMP$_s$ is correlated with lifespan in WT strains. (A)** Internal potassium as a function of external potassium. **(B)** MMP$_s$ as a function of internal potassium. **(C)** Median replicative lifespan as a function of MMP$_s$. MMP$_s$ was normalized to the WT strain grown in media with standard potassium and arginine as the nitrogen source. Shapes indicate the amount of external potassium. $r$ indicates the Pearson correlation coefficient, and $p$ is the P-value of the estimate. Internal potassium data are from same experiments used to generate Fig. 4 A. MMP$_s$ data are from same experiments shown in Fig. 5 C. Median lifespan data are from same experiments shown in Fig. 5 B.

buffer containing 58 mM KCl had higher MMP than the same mitochondria diluted into buffer containing 250 mM KCl (Fig. 7 B).

We tested whether the effect of KCl on MMP was specific to potassium by incubating isolated mitochondria in 50 mM KCl plus either 200 mM choline chloride (Fig. 7 C, purple) or 200 mM NaCl (Fig. S5 J, purple) and found both salts reduced MMP as much as the 250 mM KCl condition (Fig. 7 C, orange; and Fig. S5 J, orange). This was not due to changes in osmolarity, as adding 400 mM sorbitol to 50 mM KCl (Fig. 7 C, pink) did not change MMP relative to 50 mM KCl alone (Fig. 7 C, green). Thus, MMP was sensitive to the ionic strength of the buffer, not the particular ion or the osmolarity of the buffer.

The effect of ionic strength was observed for each of the ETC substrates and appeared to affect the substrates in a similar way (Fig. 7 D; P >0.2 for all interactions between substrate and KCl concentration), indicating that the effect of ionic strength exerted its influence downstream of the dehydrogenases (e.g., on complex III, cytochrome *c*, or complex IV) (Rosenfeld and Beauvoit 2003; Fraenkel 2011), or had a more general effect on the inner membrane, such as increased inner membrane permeability to protons (Campbell et al., 1975; Lambert and Merry 2004). Increased membrane permeability would lead to increased respiration (Nicholls 2004; Schmidt et al., 2021). However, the OCR of isolated mitochondria declined with increasing ionic strength (Fig. 7 E). This ruled out the membrane leak hypothesis and instead suggested that ionic strength interferes with overall flux through the ETC. Consistent with this hypothesis, we found that in intact, isolated mitochondria, the activity of cytochrome *c* oxidase (complex IV) decreased with increasing ionic strength, using either KCl or NaCl (Fig. 7 F).

Taken together, these experiments suggested that, *in vivo*, internal potassium levels directly affect MMP by changing the ionic strength of the cytoplasm. These changes to MMP may then contribute to overall cell health as measured by lifespan.

## Discussion

In this study, we combined *in vivo* MMP sensors with technologies we recently developed (Thayer et al., 2022, *Preprint*; Hendrickson et al., 2018) to follow the decline in MMP in aging *S. cerevisiae* cells (Hughes and Gottschling 2012). This revealed that young cells with higher MMP are more likely to have a longer lifespan than their siblings with lower MMP (Fig. 1 E). This suggests that early mitochondrial health can have a persistent impact on overall cellular health, and complements the observation made in worms that artificially boosting MMP extends lifespan (Berry et al., 2023). It is worth noting that respiratory requirements of yeast cells appear to increase with age (Leupold et al., 2019). This age-associated change in respiration is similar to the observation that the respiratory capacity of CD8[+] T cells increases as they age, though a mechanistic understanding of this change is unclear in T cells (Quinn et al., 2020).

Using the tools we developed in a genome-wide screen (Fig. S3 A), we identified gene deletion alleles that delayed or abrogated the age-associated decline in MMP (Table S2). We further characterized the allele that provided the greatest extension of lifespan, *sis2Δ* (Table S2). We found that the *sis2Δ* mutant appeared to act through its role as an inhibitor of Ppz1p activity (Fig. 2, E–G). One role of Ppz1p is destabilizing nutrient transporters through its phosphatase activity (Ariño et al., 2010; Lee et al., 2019). Consistent with this, we found that deleting the kinases *SAT4* and *HAL5*, which stabilize nutrient transporters (Mulet et al., 1999; Pérez-Valle et al., 2007; Tumolo et al., 2020), had the same effect as deleting *SIS2* (Fig. 3). We hypothesized that changes in MMP and lifespan are modulated by the activity of the high-affinity potassium transporter, Trk1p, since it is a known target of Ppz1p, Sat4p, and Hal5p (Mulet et al., 1999; Pérez-Valle et al., 2007; Tumolo et al., 2020). Consistent with this hypothesis, deleting *TRK1* extended lifespan (Sasikumar et al., 2019), and increased MMP (Fig. 4 C).

Synthesizing these data led us to a model (Fig. S5 K), whereby increasing the ability of Ppz1p to phosphorylate the high-affinity

Figure 7. **Potassium affects MMP and OCR in isolated mitochondria by altering the ionic strength of the mitochondrial environment. (A)** Effect of potassium ion (as KCl) on the MMP of isolated mitochondria, as measured by the TMRM ratio (Materials and methods). Conditions were applied sequentially to the same wells, such that the same samples of mitochondria were exposed sequentially to substrates (green), ADP (orange), and FCCP (purple). Slopes of "-adp, -fccp" (green) and "+adp, -fccp" (orange) conditions are significantly different from zero (P < 0.001 in each case). Differences between -adp, -fccp (green) and +adp, -fccp (orange) at every KCl concentration have a P-value ≤0.001. Data are from three experiments with three replicates per condition. **(B)** Isolated mitochondria were incubated in 250 mM KCl for 15 min and then switched to the indicated buffer by dilution. Data are from two experiments with three replicates per condition. **(C)** MMP of isolated mitochondria incubated in the indicated buffers. Data are from two experiments, with three replicates per condition. **(D)** Isolated mitochondria were incubated with the indicated substrate(s) in 50 or 250 mM KCl. Decline in MMP with increased KCl was equivalent for all substrates (P > 0.08 across comparisons of substrate by KCl interaction terms). Data are from three experiments, with three replicates per condition. In A–D, to facilitate comparisons across experiments, TMRM ratios were normalized to the condition containing all substrates, 250 mM KCl, and no FCCP or ADP. **(E)** OCR of isolated mitochondria incubated in the indicated amount of potassium. Data are from two experiments, with 15 replicates per condition. **(F)** Cytochrome c oxidase activity as a function of ionic strength in milliequivalents (mEq.) Ionic strength was altered using KCl (green) or 40 mM KCl plus different amounts of NaCl (orange) to obtain the indicated ionic strength. Data are from four experiments, with 3–4 replicates per experiment. All error bars show 95% CIs.

potassium transporter, Trk1p, by eliminating either its inhibitor, Sis2p, or the kinases Sat4p and/or Hal5p, reduced the activity of Trk1p, leading to lower internal potassium, higher MMP, and longer lifespan. Indeed, reducing environmental potassium was sufficient to extend lifespan (as observed before [Sasikumar et al., 2019]), and increase the MMP of a WT strain to the same levels as a sis2Δ mutant grown in a standard potassium environment (Fig. 5, A and B). Interestingly, sis2Δ was longer-lived at all external potassium concentrations, suggesting that deleting SIS2 has other lifespan-extending effects in addition to limiting internal potassium. Transcriptional (Table S4) and metabolic (Table S3) evidence suggests this may be related to a chronic reduction in amino acid levels, which has been shown to extend lifespan in yeast, flies, and mammals (Orentreich et al., 1993; Johnson and Johnson 2014; Grandison et al., 2009; Fulton et al., 2024). Although vacuolar pH did not appear to explain the young cell phenotypic differences between WT and sis2Δ (Fig. S4 E), we

note that vacuolar pH plays a role in aging more generally (Henderson et al., 2014; Antenor et al., 2025), and could further affect aging phenotypes.

Nitrogen source and genotype affected potassium homeostasis by modulating the relationship between external and internal potassium, but, ultimately, MMP was determined by the concentration of internal cellular potassium (Fig. 6 B). Using isolated mitochondria, we determined that this was a direct effect of potassium through its influence on the ionic strength of the mitochondrial environment (Fig. 7). Because ionic strength is determined by charge density and not the identity of the ion, in theory any cation could have a similar effect on mitochondrial function. However, since potassium is the most abundant cation in the cell by an order of magnitude (Satlin 2009; Mulkidjanian et al., 2012), it largely determines the ionic strength of the cytoplasm and the mitochondrial inner membrane space to which the mitochondrial inner membrane is exposed (Cortese et al., 1991).

Mechanistically, our experiments appear to rule out an effect of ionic strength on the ATP synthase (complex V) (Fig. 7 A), the dehydrogenases before complex IV (Fig. 7 D), and changes in membrane permeability (Fig. 7 E). Rather, we provide evidence that complex IV activity is sensitive to ionic strength (Fig. 7 F). This mirrors results in other systems (from beef heart and rat liver), where complex IV activity has been shown to be sensitive to ionic strength, both by directly modulating its interaction with (Wilms et al., 1980; Wilms et al., 1981) and by altering the diffusivity of (Gupte and Hackenbrock 1988) cytochrome *c*. As we show here, we can attribute reductions in MMP above ~120 mM potassium to reduced activity of complex IV, but the further increase of MMP observed below 120 mM may be due to the modulation of another factor, or a limitation related to determining complex IV activity in isolated mitochondria.

Potassium is the most abundant intracellular cation across all domains of life (Stautz et al., 2021; Ariño et al., 2010; Satlin 2009; Vašák and Schnabl 2016), and is maintained at intracellular concentrations in the hundreds of millimolar—much higher than any other cation (Satlin 2009; Mulkidjanian et al., 2012). This reflects the highly conserved role of potassium as a critical determinant of fundamental cellular attributes, including cell volume, cytoplasmic pH, and plasma membrane potential (Ariño et al., 2010). Here, we suggest that MMP should be added to this list.

Cells maintain internal potassium within a relatively narrow range, but this range is correlated with external potassium in both yeast (Fig. 6 A, [Kahm et al., 2012; Herrera et al., 2014; Masaryk and Sychrová 2022]) and mammalian cells (Patrick 1978; Relman et al., 1961; Sterns et al., 1981). Regulation of intracellular potassium levels in an environment of fluctuating extracellular potassium is certainly relevant to microorganisms, but it is also important in mammalian tissues, such as the gut epithelium, the liver, and the kidney, which are exposed to widely varying levels of potassium (McDonough and Youn 2017; Satlin 2009). Cells of the immune system are also exposed to a wide variety of extracellular potassium concentrations, with potentially negative consequences. For instance, the high potassium tumor microenvironment can lead to suppression of T cell effector functions and their ability to differentiate (Vodnala et al., 2019). Intriguingly, this is associated with reduced MMP (Eil et al., 2016).

Loss of homeostatic balance in cellular systems is a principal characteristic of aging in every organism (Pomatto and Davies 2017). In humans, the ability to maintain potassium homeostasis declines with age, leading to a total body reduction of potassium content (as measured by isotope labeling), thought to be mostly due to loss of muscle mass (Andreucci et al., 1996; Cox and Shalaby 1981). Intracellularly, red blood cell potassium content also declines, but it is not known whether this is true of other cell types (Andreucci et al., 1996). Clearly, the cause-and-effect relationships between aging, potassium homeostasis, and mitochondrial function have not been worked out. More generally, failure to maintain ion homeostasis can lead to aging-related diseases, including channelopathies in cancer (Pi et al., 2007; Prevarskaya et al., 2018), defects in insulin signaling (De Marchi et al., 2021), Alzheimer's disease (Song et al., 2024), and Parkinson's disease (Zhuang et al., 2024).

In this study, we revealed a crucial role for potassium ion homeostasis in controlling and maintaining mitochondrial function as cells age, and we show that one way to intervene is through manipulation of the activity of potassium transport. There are a wide variety of small molecules that can be used to manipulate the activity of mammalian mitochondrial potassium channels, but their effects on membrane potential and longevity have not been investigated (Wrzosek et al., 2020). Based on our results, we believe this approach could provide new potential avenues for slowing the age-associated decline in mitochondrial function and, as a consequence, slow the onset of aging-related disease.

## Materials and methods

### Strain construction

Reporter strains were derived from BY4700 (MATa *ura3Δ0*) (Baker et al. 1998). Nonreporter strains were derived from a cross between BY4707 (MATα *met15Δ0*) and BY4717 (MATα *ade2Δ0::hisG*). We found that BY4707 and BY4717 contain a null allele of *PTR3* due to insertion of a TY element. Since *ptr3⁻* alleles extend lifespan (Tsang et al., 2015) and are synthetic-lethal with *leu⁻*, we corrected this defect in the following way. A prototrophic strain derived from a cross between BY4707 and BY4717 (CGY2.66) was crossed with BY4713 (MATα *leu2Δ0*), and a *leu⁻* haploid was selected. A prototrophic, *PTR3⁺ LEU⁺* strain (CGY4.50) was generated by PCR amplification of full-length *LEU* and transformation followed by selection on plates lacking leucine. The genotype was confirmed by whole-genome sequencing.

To construct the reporter strain, a plasmid containing the mCherry coding sequence and natMX selection cassette (CGB4.09) was PCR-amplified using primers with homology to the C-terminal region of *TIM50* and introduced into BY4700 by transformation. ClonNat-resistant colonies were selected and sequence-verified. The resulting strain was transformed with NotI-digested plasmid containing pTPI-preCOX4-SL17-mNeon-Green-URA3 (CGB2.36), which has homology to a gene-free region of chromosome I (Hughes and Gottschling 2012). The sequence of the preCOX4 leader was 5′-ATGCTTTCACTACGTC AATCTATAAGATTTTTCAAGCCAGCCACAAGAACTTTGTGT AGCTCTAG-3′. The sequence of the SL17 degron was 5′-TCGAT TAGTTTCGTAATACGTTCACATGCTTCGATTAGAATGGGGGCA TCCAATGATTTTTTCCACAAGTTGTACTTCACGAAATGCCTC ACATCAGTTATTTTATCTAAATTTCTGATC-3′.

We confirmed that the expression of both *TPI1* and *TIM50* did not change after 24 h of aging ($\log_2$[old/young] = −0.07, P = 0.48; and $\log_2$[old/young] = −0.1, P = 0.12, respectively; Table S1) and that reporter expression had a negligible effect on lifespan (mean lifespan 25.6 divisions without reporter vs 24.2 divisions with reporter, P = 0.003; Fig. S2 C).

Gene deletions were made using standard techniques, either by amplifying the indicated drug resistance marker from a plasmid with primers containing homology to the 5′- and 3′-regions immediately surrounding the ORF, or by amplifying the deletion cassette containing kanamycin resistance out of the library strain. Gene deletions were confirmed using PCR, and all

strains were checked for functional mitochondria by growth on 2% agar plates containing 1.0% yeast extract, 2.0% peptone, 2.0% ethanol, and 3.0% glycerol. In general, at least two independent clones were analyzed. A full list of strains can be found in Table S5, and oligos can be found in Table S6.

## General media and growth conditions
Unless indicated otherwise, cells for all assays were prepared in the following way. Strains to be assayed were streaked from a −80°C freezer onto 2% agar plates containing 1% yeast extract, 2% peptone, 2% glucose and grown until colonies were visible (typically 2–3 days). Cells were inoculated into synthetic complete (SC) medium (2 g 10X SC mix [Sunrise], 6.7 g yeast nitrogen base (YNB) containing ammonium sulfate [Sunrise], 20 g glucose per liter, buffered to pH 4.5, filter-sterilized) and grown for 16–24 h to saturation. These "overnight" cultures were then diluted into the experimental media and grown for 24 h in the exponential phase, diluting as necessary so that the cell culture never exceeded an $OD_{600}$ of 0.1. For potassium titration experiments, we used a custom formulation of "Translucent K+ free medium" (Formedium), with 960 mg/l (8 mM) sodium dihydrogen phosphate instead of 920 mg/l (8 mM) ammonium dihydrogen phosphate. To this, either 5 g/l (38 mM) ammonium sulfate or 2.1 g/l (10 mM) arginine HCl was added as a nitrogen source.

## Mitochondrial isolation
Mitochondria were isolated by the fractionation and differential centrifugation method (Albers et al., 2006; Liao et al., 2018). Specifically, cells were inoculated into SC media and grown for 16–24 h to saturation ($OD_{600}$ ~4). The $OD_{600}$ of the culture was measured just prior to sampling. 400 $OD_{600}$ of cells were collected on a filter and washed with water. Cells were resuspended in 35 ml water and spun down at 1,500 $g$ for 10 min, and the supernatant was removed. Cells were resuspended in 20 ml cell wall destabilization buffer (100 mM Tris-HCl, pH 9.5, 50 mM β-mercaptoethanol) and incubated at 30°C for 15 min. Cells were then washed with 20 ml sorbitol buffer (1.2 M sorbitol, 5 mM $MgCl_2$, 20 mM potassium phosphate, pH 7.5) and digested in 4 ml sorbitol buffer with 12 μl 200 mg/ml 100T zymolyase at 30°C for 90 min. Digested cells were washed with 20 ml ice-cold sorbitol buffer and resuspended in 4 ml ice-cold mitochondrial isolation buffer (0.6 M sorbitol, 20 mM HEPES, pH 7.5) to which protease inhibitor (cOmplete protease inhibitor cocktail, Roche) was added. Cells were lysed with a Dounce homogenizer using 15 strokes with the tight pestle (Kimble Kontes 2 ml Dounce Tissue Grinder, DWK Life Sciences). Lysed cells were collected, and 4 ml ice-cold mitochondrial isolation buffer with protease inhibitor was added. Lysed cells were spun down at 3,000 $g$ for 5 min. The supernatant was collected and spun again at 12,000 $g$ for 10 min. The supernatant was discarded, and the mitochondrial pellet was resuspended in ice-cold mitochondrial isolation buffer with protease inhibitor. The mitochondrial concentration was quantified using a Pierce BCA protein assay (Thermo Fisher Scientific). All assays on isolated mitochondria were performed in mitochondrial assay solution (MAS) containing 220 mM mannitol, 70 mM sucrose, 10 mM $NaH_2PO_4$ (dihydrate), 5 mM $MgCl_2$, 2 mM HEPES, 1 mM EGTA, 0.2% (w/v) fatty acid–free BSA.

## MMP aging screen
We crossed the strain containing the MMP reporter system into the yeast knockout collection (Winzeler et al., 1999), in which each of the ~4,700 nonessential genes has been replaced with a unique barcode. This deletion library was pooled and aged using MADs (Hendrickson et al., 2018) for 24 h, and the protocol was adapted as follows. Cells were grown for 24 h at low $OD_{600}$ (<0.05) in SC medium containing 2% methyl-alpha-D-mannopyranoside (MP, Sigma-Aldrich) at pH 4.5. The presence of MP in the media prevents age-related aggregation of cells in the MAD (Hotz et al., 2022), similar to mannose used in the original protocol. For cell beading, one tube of Super Mag NHS-activated 100-nm beads (Ocean NanoTech) was equilibrated to room temperature for 30 min and then resuspended in 400 μl resuspension buffer (25 mM MES, pH 6.0, 0.01% Tween-20). Then, beads were vortexed at 1,600 rpm until needed. Four ODs of cells per MAD were harvested, washed extensively in labeling medium (YNB, 2% glucose, 2% MP, pH 7.3), and resuspended in 400 μl labeling medium. Magnetic beads were placed on a magnet, resuspension buffer was taken off, 200 μl of labeling medium was added, and 40 μl of beads was immediately mixed with each aliquot of resuspended cells. 600 μl PEG-3000 was added, and the tubes were incubated at room temperature on a rotating wheel for 15 min. Then, cells were washed twice on a magnet with growth medium and the quality and uniformity of beading were checked under a microscope.

MADs were loaded in triplicate and run for 24 h ("T24"). After harvesting, cells were immediately fixed in 100 mM Tris base, 20 mM sodium azide, 20 mM sodium fluoride, adjusted to pH 7.6 (TAF [Olmo and Grote 2010]) buffer to prevent further changes to the MMP ratio, and then washed several times on magnets to remove daughter cells. As a control, we used mother cells from an unaged population, which was bead-labeled as above and harvested after 3 h ("T3") of growth in SC medium. The resulting aged and unaged samples were stained with wheat germ agglutinin (WGA) conjugated to a 647-nm Alexa Fluor (Thermo Fisher Scientific; WGA647) to mark bud scars, and SYTOX Blue (Thermo Fisher Scientific) to mark dead cells. These enriched, stained populations were sorted on a BD FACSAria (BD Biosciences). To further purify the mother cell population and remove dead cells, only cells with high WGA, low SYTOX Blue signals were considered for analysis (Fig. S3 B). From the resulting live mother population, the distribution of the mNeon-Green/mCherry ratio was divided into "low," "medium," and "high" populations, and $10^5$–$10^6$ cells were collected for each subpopulation. In total, 10 sets of populations were collected for young and old mother cells.

From these populations, barcode libraries were generated in 2 or 3 technical replicates as follows. Genomic DNA was extracted using YeaStar Genomic DNA Kit (Zymo Research), and DNA yield was quantified using the Qubit dsDNA High Sensitivity kit (Invitrogen). Barcodes were amplified by PCR using 10-ng genomic DNA and custom oligos containing Illumina indices (98°C for 1 min; 15 cycles of 98°C for 20 s, 66°C for 30 s, 72°C for 30 s; 72°C for 2 min). The PCR product was purified with RNAClean XP beads (Beckman Coulter) and run on Agilent Bioanalyzer for quantification and quality control. Samples were

pooled in equal concentrations and sequenced on an Illumina HiSeq 2000.

The entire screen was performed twice with slight differences in the protocol. In the first experiment, the low, medium, and high populations were defined as being below the T3 mode, above the T3 mode, and the top 25% of the T3 population. In the second experiment, each population distribution was split into thirds to make the low, medium, and high groups. The method of sorting had very little influence on the results, as shown by the fact that both the mean counts (Fig. S3 D) and significant log fold changes (Fig. S3 E) were highly correlated between the two experiments. We used the second experiment for further analysis.

### Screen analysis
#### Barcode detection algorithm
To identify barcodes with high recovery and high accuracy, we developed an algorithm that utilized information from both the sense and antisense strands of the paired-end reads. This algorithm takes advantage of the additional information available from paired-end reads. First, the common sequence outside of the (variable) barcode is identified in the sense and antisense direction using regular expression matching and allowing up to two mismatches on either side of the barcode. This allowed us to identify 98% of reads in at least one direction. Once the common sequence was identified, the associated barcode was extracted. If the barcode was present in both directions and the sequences were identical, or if the barcode was only identified in the sense or antisense direction, we used that sequence as the barcode. If the barcode was present in the sense and antisense direction, but the barcode sequences did not agree, we chose the bases with the highest associated quality score to create the barcode. Using this technique, we recovered a barcode in at least one direction from >87% of sequences.

Next, the cosine similarity of each extracted barcode to the set of known barcodes in the collection was calculated, and the highest scoring member from the barcode collection was selected as the match. Thus 300000, the cosine similarity score was also the confidence in the match. Using this approach, we did not have to fix a number of mismatches, insertions, or deletions to be considered a match. This approach also had the advantage of correctly identifying barcodes even if there were errors in the published barcode sequences associated with the collection. Empirically, we determined a minimum cosine similarity score of 0.8 (with 1 being a perfect match) was a conservative threshold, and this included 96% of the extracted barcodes.

#### Differential barcode enrichment
Differential enrichment of barcodes in the high vs low mNeonGreen/mCherry ratio groups was analyzed using the R package "DESeq2" (Love et al., 2014). Log fold change estimates were shrunk using the R package "apeglm" (Zhu et al., 2019).

### Statistical modeling
Unless otherwise indicated, statistical modeling was performed using the functions "lm" from base R (R Core Team 2024) for fixed-effect models, "lmer" from the "lme4" package for mixed-

effect models (Bates et al., 2015), or "coxph" from the "survival" package (Therneau and Grambsch 2000) for lifespan comparisons. The specific formulas used for each model are referenced in their corresponding sections below.

Point estimates, contrasts, and P-values (all adjusted for multiple hypothesis testing) were estimated using the R package "emmeans" (Lenth 2024), using ordinary marginal means and the default P-value adjustment method "tukey." Correlations were calculated using the R function "cor.test" in base R. Datasets whose natural range was only positive values (e.g., sizes, fluorescence intensities, concentrations) were log-transformed before analyzing to improve normality and satisfy the assumptions of linear modeling (i.e., that data span the range of negative to positive infinity). Other datasets were assumed to be normally distributed, but this was not formally tested.

### Yeast Lifespan Machine
The YLMs were operated according to the freely available documentation (Thayer et al., 2022, *Preprint*). All experiments were performed at 30°C. Bright-field images were taken every 15 min. Remade mutants from the screen were run on a DMi8 (Leica) microscope, and fluorescent images were taken every 2 h using a SpectraX (Lumencore) light source, a "GFP" filter cube (470/40-nm excitation, 495-nm dichroic, 525/50-nm emission), and a "Texas Red" filter cube (560/40-nm excitation, 585-nm dichroic, 630/75-nm emission) for mNeonGreen and mCherry, respectively, a 40× air objective (Leica, 0.95 NA), and a pco.edge 4.2 CLHS sCMOS camera (Excelitas). Each gene deletion was represented by at least two clones, and each clone was analyzed in at least two microfluidic channels within one experiment. Some clone replicates were lost due to channel clogging or other technical issues. The WT strain was included in every experiment as a reference.

Subsequent lifespans were acquired on custom-made microscopes (Thayer et al., 2022, *Preprint*) equipped with 40× air objectives (Nikon, 0.95 NA) and pco.panda 4.2 sCMOS (Excelitas) cameras, and using microfluidics devices with "DetecDiv"-style catchers (Aspert et al., 2022). No fluorescence illumination was used. Every clone was analyzed in at least two channels within one experiment, and at least two experiments were performed. Experiments were run for 99 h. Channels that were observed to clog with cells before 66 h were excluded from further analysis.

#### MMP trajectory analysis
Fluorescence data were extracted using custom software. Raw trajectory data were filtered to only include trajectories that started at time 0, stopped before time point 300 (to avoid false event calls at the end of the run), and had $MMP_s$ and $MMP_i$ signals >50. Raw trajectory data are available in Table S7.

MMP ratio comparisons with WT after 24 h were performed by extracting MMP ratio data from cells that were between 21.5 and 26.5 h old. The data were then fit to a statistical model with a term for the interaction between genotype and time, a random effect for the trajectory id, and a random effect for the experiment: "$\log10(MMP\ ratio) \sim gene * time + (1|id\_lifespan) + (1|experiment)$." Individual clones and microfluidic channels

were not explicitly modeled in order to average over their contributions.

## Lifespan analysis

The genomes of clones used in lifespan analysis were sequenced to ensure that rDNA copy number was not too low (≥100 copies [Hotz et al., 2022]). Bud counts were determined from a machine learning model (Thayer et al., 2022, *Preprint*) using publicly available code (https://github.com/calico/ylmcv). Lifespan comparisons and estimates of the median replicative lifespan were made with Cox proportional hazard (CPH) models using the "survival" package in R (Therneau and Grambsch 2000). The basic model used was "Surv(bud count, event) ~ gene," where "event" was a parameter describing whether the cell was observed to die in the catcher, or washed out at some point (i.e., censored from the data). The experiment and channel information was concatenated to make a unique identifier and used as the clustering parameter to account for within-channel correlations. Potassium concentration and nitrogen source type were added as interacting terms with genotype as appropriate.

## Culture doubling time

Culture doubling time was calculated over the 24-h period of exponential growth preceding every experiment. These data were fit to a model of the form "log10(doubling time) ~ gene + experiment."

## Mitochondrial OCR

OCR was measured on a Seahorse XFe96 (Agilent) according to the manufacturer's instructions. All temperature-sensitive steps were performed at 30°C.

### Intact cells

Each cell sample was diluted to an $OD_{600}$ of 0.1, and 180 μl was added to the cell culture plate. The plate was then spun down for 3 min at 500 rpm. 20 μl antimycin A (AA) was added to port A at a concentration of 25 μM. Three measurements of the OCR were made. AA was injected (final concentration 2.5 μM in growth media) to inhibit complex III, and then three more measurements of OCR were made. The measurements before and after adding AA were averaged over time, replicate, and strain. The mitochondrial OCR was calculated as the difference between the averaged OCR before adding AA and the averaged OCR after adding AA.

### Isolated mitochondria

Isolated mitochondria were added to a final concentration of 0.12 mg/ml in 180 μl MAS plus the indicated concentration of KCl. An injection solution of 5 μM AA was prepared in MAS buffer plus the appropriate concentration of KCl. After three baseline measurements, 20 μl of the injection solution was added to the well (0.5 μM final concentration) and three more measurements were made. Mitochondrial OCR was calculated as above. Data were analyzed across experiments using two linear models. The first provided point estimates at each potassium concentration by treating potassium concentration as a factor, while the second provided the intercept and slope of the line

through the points by treating potassium concentration as a continuous variable. In both cases, the formula was "mitochondrial_OCR ~ potassium + experiment," where "potassium" was the potassium concentration, either treated as a factor or a continuous variable.

## Assays of mitochondrial membrane potential

### Protein-based reporter

Cells were grown as described in "General media and growth conditions" in the indicated media. Cells were collected to a final $OD_{600}$ of ~0.01, and TAF was added to stop metabolic function (Olmo and Grote 2010). SYTOX Blue (Thermo Fisher Scientific) was added for live/dead cell staining. For every sample, at least 10,000 cells were analyzed on BD LSRFortessa X-20 (BD Biosciences) using the FITC (488-nm excitation, 505 long-pass (LP), 525/50-nm emission) and PE-Texas Red (561-nm excitation, 595LP, 610/20-nm emission) filter sets to measure mNeonGreen and mCherry signals, respectively. SYTOX Blue was measured using the BV421 (408-nm excitation, 450/50-nm emission) filter set. At least three technical replicates were collected for each sample. At least two experiments were performed on different days to generate the data for each figure.

The mean values for each channel and the mean value for the ratio of the green to red channels were fit separately using the formula "log10(signal) ~ gene + experiment," where "experiment" was added as a factor to account for batch effects. The effects of concentration and nitrogen source were modeled as interacting terms with genotype, such that the most complicated model involving multiple genotypes, nitrogen sources, and potassium concentrations was "log10(signal) ~ gene * concentration * nitrogen + experiment."

### General TMRM staining protocol

TMRM staining was performed similar to (Hughes and Gottschling 2012), with some modifications. Cells were grown in the exponential phase for 24 h to an $OD_{600}$ < 0.1. Cells were sampled to a final total $OD_{600}$ of 0.02 (~500,000 cells) into 1.5-ml tubes. To each tube, 5 μl of 50% polyethylene glycol 3,350 (PEG) was added to assist in forming a cell pellet upon spinning. Cells were spun down at 6,500 $g$ for 1 min, the supernatant was removed, and cells were resuspended in 500 μl HEPES buffer containing 5% glucose (10 mM HEPES, 5% glucose, pH 7.6). Cells were spun at 6,500 $g$ for 30 s, and the supernatant was removed. Cells were resuspended in 500 μl TMRM buffer (50 nM TMRM, 10 mM HEPES, 0.25% PEG) and incubated in the dark at room temperature for 15 min. Cells were spun down and resuspended in 600 μl HEPES buffer (without PEG) containing 6 μl SYTOX Blue for live/dead staining.

### TMRM assay by flow cytometry

Stained cells were analyzed on BD LSRFortessa X-20 (BD Biosciences) using the PE channel (561-nm excitation, 586/15-nm emission). The TMRM signal decayed exponentially during the course of analysis, a result of dye export from the cell (Homolya et al., 1993); this decay rate varied depending on the mutant being analyzed. To correct for this decay, we measured each set of samples at least three times, analyzing 10,000 cells for each

sample. We then calculated the exponential decay rate constant to correct the signal. Experiments were performed at least twice on different days. Results were analyzed using the corrected mean values of each sample using the formula "log10(TMRM) ~ gene + experiment," with experiment as a factor to account for batch effects.

For the comparison of TMRM with the MMP$_s$ reporter in young cells (Fig. S3 F), strains containing the reporter system were grown to exponential phase, stained, and analyzed by flow cytometry as above. (TMRM is more than 10-fold brighter than the TIM50-mCherry signal, so the contribution of mCherry could be ignored.) The mean values of the corrected TMRM signal and the mNeonGreen signal (from the FITC channel) for each strain were analyzed using the model formula "log10(signal) ~ gene + experiment." The resulting estimates of the MMP$_s$ signal (on the original, linear scale) were adjusted in the following way. For each deletion mutant, the mitochondrial mass adjustment was estimated by taking the fractional change, $d$, of MMP$_i$ compared with WT using data from the YLM experiments by averaging over cells 0–5 divisions old. Then, that mutant's corresponding MMP$_s$ value was multiplied by $-(1 + d)$ to get the adjusted estimate. The Deming regression estimate was made using the R package "deming" (Therneau 2024).

### TMRM ratio of isolated mitochondria

Isolated mitochondria were resuspended to the indicated concentration (0.3–0.5 mg/ml) in MAS plus the indicated concentration of KCl. Each condition and replicate were paired, with TMRM added to one "signal" well to a final concentration of 200 nM, and the equivalent volume of DMSO added to the other "background" well. Unless otherwise indicated, 10 mM glutamate, 10 mM malate, 10 mM succinate, and 10 mM glycerol-3-phosphate were added as substrates. The final volume used in analysis was 100 μl. TMRM was measured ratiometrically by monitoring emission at 605 nm while exciting at 546 or 572 nm (Scaduto and Grotyohann 1999) in a CLARIOstar Plus plate reader (BMG Labtech). For each experiment, conditions were performed in triplicate, and five sequential measurements were made to improve estimates of the signal. If required, ADP was added to a final concentration of 0.5 mM, and FCCP was added to a final concentration of 15 μM. For experiments that used ADP and/or FCCP, the plate was first read without ADP or FCCP. Then, ADP (if required) was added to each well and the plate was read again. Finally, FCCP was added to each well and the plate read again. Background for each wavelength was calculated by averaging over the background wells for each condition. After subtracting the averaged background, the 572/546 ratio was calculated for each well and each time point, and then data from the individual time points were averaged to give the final result.

Analysis was performed using the model "TMRM ratio ~ buffer * treatment + experiment," where "buffer" indicated which buffer the mitochondria were incubated in, and "treatment" indicated which combination of ADP and FCCP had been added. For experiments where determining the slope was important, data were analyzed using two linear models. The first provided point estimates at each potassium concentration by treating the buffer as a factor, while the second provided the

intercepts and slopes of the lines through the points by treating the buffer as a continuous variable. When the same well was measured multiple times in an experiment, a random effect for the well was added to account for the nonindependence of these measurements.

### Kinetics comparison of TMRM and MMP$_s$ construct

A strain (CGY62.43; Table S5) containing the MMP$_s$ part of the construct was grown as described above (General media and growth conditions), and a total of 0.02 OD$_{600}$ of cells were stained with TMRM as described above (TMRM staining protocol) with the following changes. After washing with 1 ml HEPES +5% glucose buffer, cells were resuspended in the same buffer plus 5 μg/ml WGA647 and incubated in the dark at room temperature for 5 min. After another wash to remove unbound WGA647, cells were resuspended in the same buffer plus 50 nM TMRM and incubated in the dark at room temperature for 45 min. From there, 150 μl was added to a glass-bottom well (18-well μ-Slide, Ibidi USA) that had been pretreated with 5 mg/ml concanavalin A in PBS. The plate was spun at 100 $g$ for 2 min to settle cells. The entire viewable portion of the well (~15 × 15 fields of view) was imaged on a microscope (Leica) with a 63× oil objective (HC PL APO, 1.40 mm NA), and kept at 30°C using an Okolab incubation system. Each field of view was imaged over a 10-μm z-stack, with a 0.5-μm step size. The filters were 470-nm excitation/510-nm emission for mNeon, 550 ex./590 em. for TMRM, and 640 ex./666–724 em. for WGA. Leica Thunder background subtraction was applied to all images.

For segmentation, four-channel 3D image volumes were processed as CxZxYxX, with bright-field in channel 0, WGA in channel 1, TMRM in channel 2, and mNeon in channel 3. Voxel sizes in z, y, and x were taken from acquisition metadata and used for all anisotropy corrections and volumetric calculations. Cells were segmented in 2D from the medial bright-field z-slice using CellposeSAM on a GPU with inverted contrast and a batch size of 64 (Pachitariu et al., 2025, Preprint). Connected components with area <300 pixels were removed. Any label touching the image border was excluded. The resulting 2D label map defined cell IDs and was broadcast across z to constrain downstream operations. For WGA and mNeon, organelle structures were segmented in 3D using Nellie's segmentation pipeline (Lefebvre et al., 2025). In brief, segmentation was performed independently within each cell footprint using a multiscale, anisotropy-aware vesselness filter, with spatial scales of 1.0 and 1.5 pixels and a z/y scale ratio set by the recorded voxel sizes. For each cell label, the distribution of positive vesselness values across z was used to compute a label-specific "Minotri" threshold (the minimum of the Otsu and Triangle thresholds (Lefebvre et al., 2025)); voxels meeting or exceeding this threshold were provisionally assigned to that label. The binary mask was cleaned with a single 3D binary closing using a spherical structuring element of radius 2 voxels and relabeled, and components smaller than 7 voxels were discarded. Surviving voxels were re-encoded by multiplying with the broadcast 2D cell label map so that every organelle voxel carried its parent cell ID. For validation, a three-plane stack per field was saved comprising the WGA mask, the mNeon mask, and the z-broadcast cell labels. Per-cell

metrics were computed from the 2D bright-field labels and the 3D organelle masks. Cell planar area (pixels) was converted to an effective radius using the in-plane pixel size, and cell volume was approximated as a sphere with that radius. Median WGA intensities were measured from the WGA image at the WGA-mask voxels. Median TMRM and mNeon intensities were measured from their respective images at the mitochondrial-mask voxels. All per-cell measurements were written to CSV files, and raw volumes and label stacks were saved as TIFFs for auditing and quality control. Implementation details and exact parameter values are available in the public code repository (https://github.com/aelefebv/nellie).

Bud scars of 2,576 cells were counted by hand and matched to existing labels from the processed images. To increase the proportion of aged cells, the data were binned on the WGA signal, and more cells were chosen from higher WGA bins for bud scar counting (Fig. S2 A). After trimming the top and bottom 1 percentile of intensity values and filtering on cell size (between 39 and 235 μm³) to exclude developing cells still connected to their mother cells, the processed data from each field of view were normalized to the average value in that field to account for experiment-, position-, time-, and density-dependent differences in intensities. The resulting normalized intensities of the TMRM and mNeon signals were fit using the model formula "log10-(intensity) ~ bud_scar * fluorophore+experiment + (1|cell_id)," where "bud_scar" indicated the number of bud scars (treated as a factor), "fluorophore" indicated the TMRM or mNeon channel, and "cell_id" uniquely described each cell. A random effect for cell_id was necessary since the intensity information for TMRM and mNeon came from measurements of the same cell. Means and CIs were determined on the original (linear) scale using emmeans. Finally, each fluorophore was normalized to its maximum value across bud scars to facilitate direct comparison of signal as a function of age.

## Plasma membrane potential

Cells were grown as above, and 1 ml of culture was sampled into a 1.5-ml tube. Di-8-ANEPPS (Invitrogen) was added to a final concentration of 1.5 μM, and 1 μl SYTOX Blue was added. Samples were incubated at room temperature in the dark for 7 min and then run on BD LSRFortessa X-20 (BD Biosciences) using the BV510 (408-nm excitation, 505LP, 525/50-nm emission) and PerCP-Cy5.5 (488-nm excitation, 635LP, 695/40-nm emission) channels. The ANEPPS ratio was calculated by dividing the PerCP-Cy5.5 signal by the BV510 signal. As with TMRM (see "TMRM assay by flow cytometry"), sample sets were set up in triplicate (or more) so that signal decay time constants could be estimated and corrected for.

Data were analyzed as described in TMRM assay by flow cytometry. The model formula used was "log10(ANEPPS ratio) ~ gene + experiment."

## Vacuolar pH

Relative vacuolar pH was measured using the vSEP ratiometric reporter construct (Okreglak et al., 2023). The plasmid containing the ratiometric vSEP construct (CGB8.15) was digested with NotI and integrated into the HO locus. After adding SYTOX

Blue, samples were analyzed on BD LSRFortessa X-20 (BD Biosciences) using the FITC (488-nm excitation, 505LP, 525/50-nm emission) and PE-Texas Red (561-nm excitation, 595LP, 610/20-nm emission) filter sets to measure vSEP ("pH-sensitive") and mCherry ("pH-insensitive") signals, respectively. SYTOX Blue was measured using the BV421 (408-nm excitation, 450/50-nm emission) filter set.

Data were analyzed as described in TMRM assay by flow cytometry, with separate models being fit to the vSEP signal, the mCherry signal, and vSEP/mCherry ratio. The model used was "log10(vSEP ratio) ~ gene * conc + experiment," where "conc" was the concentration of KCl in the media.

## RNAseq

For the young/old cell comparison, cells were grown in exponential culture or aged in MADs for 24 h as above (MMP aging screen) and harvested in RNA*later* (Sigma-Aldrich). Bulk RNA-seq was performed by Admera Health. Total RNA was extracted and purified using RNeasy Plant Mini Kit for RNA Extraction (Qiagen). RNA was quality-controlled and quantified using a TapeStation (Agilent) and only high-quality samples (RNA integrity number >6) were used. NEB Ultra II Directional RNA Library Prep Kit (New England Biolabs Inc.) was used to generate cDNA and sequencing libraries. Libraries were indexed using Admera's in-house index primers and cleaned with AFT-Mag NGS DNA Clean Beads (ABclonal). The indexed libraries were pooled and sequenced on the Illumina NovaSeq X plus platform with a read length of 2 × 150 bp. Raw sequencing reads were demultiplexed with bcl2fastq.

For the comparison of *sis2*Δ and WT in standard (7.35 mM) or low (0.5 mM) KCl, strains were grown to saturation overnight in media containing standard potassium. From these overnight cultures, strains were diluted into media containing the appropriate potassium concentration in biological triplicate and grown for an additional 24 h, ensuring that the $OD_{600}$ did not go above ~0.1. The $OD_{600}$ of each culture was measured just prior to sampling. Enough volume to obtain a total of 0.5 $OD_{600}$ of cells was applied to a vacuum filtration system onto a 0.45-μm nylon filter (HNWP02500, MilliporeSigma). Cell pellets were resuspended in 200 μl of lysis buffer (10 mM Tris [pH 8.0], 0.5% SDS, 10 mM EDTA). After adding 200 μl acid phenol (pH 4.3), samples were vortexed for 30 s. Samples were then incubated at 65°C for 30 min in a thermomixer with intermittent shaking (2,000 rpm for 1 min every 15 min). Finally, a total of 400 μl of ethanol was added and the RNA was purified using the Direct-zol RNA Miniprep Plus kit (Zymo Research) according to the manufacturer's instructions, including the optional DNase digestion step. RNA integrity was confirmed using Agilent Bioanalyzer.

Differential gene expression was analyzed using the R package DESeq2 (Love et al., 2014). Strain and potassium condition information was concatenated and used as the covariate in the analysis. Log fold change shrinkage was performed with the "ashr" R package (Stephens, 2017). The R package "clusterPro-filer" (Yu, 2024) was used to perform gene set enrichment. Enrichment was performed with the "enrichGO" function, and redundant GO terms were removed using the "simplify" function with a significance cutoff of 0.5.

## Metabolomics

Strains were inoculated in biological triplicate for each experiment. The $OD_{600}$ of each culture was measured just prior to sampling. Enough volume to obtain a total of 2.5 $OD_{600}$ of cells was applied to a vacuum filtration system through a 0.5-μm nylon filter (HNWP02500, MilliporeSigma). Using forceps, the filter was immediately removed and submerged into 3 ml of –20°C extraction solvent (80:20 methanol:water) in a 5-ml tube. Samples were stored at –20°C for at least 20 min. On wet ice, the extraction solvent was pipetted over each filter to get cells off of the membrane, and the membrane was removed and discarded. Samples were spun at maximum speed at 4°C in a microcentrifuge for 5 min. Samples were diluted an additional 10-fold into 40:40:20 acetonitrile:methanol:water after membrane removal and centrifugation. Undiluted and diluted samples were then transferred to an autosampler vial for analysis.

Targeted metabolomics were acquired on an Agilent Bio 1290 UPLC coupled to a Sciex 7500 QTRAP. MS acquisition occurred in scheduled MRM mode with two transitions per compound whenever possible. Dwell time was set to a minimum of 4 ms and a maximum of 250 ms, with a target cycle time of 1 s. Source conditions were as follows. Ion source gas 1: 60 psi; ion source gas 2: 60 psi; curtain gas: 46 psi; CAD gas: 8; source temperature: 350°C. Polarity switching was performed with a 1,600 V spray voltage for positive mode, 1,900 V for negative mode, and a pause time of 4 ms. Chromatographic separation for both methods was achieved on a SeQuant ZIC-pHILIC 5-μm 200 Å 150 × 2.1 mm HPLC column. Mobile phase A was 20 mM ammonium carbonate, pH 9.2, with 1 μM medronic acid. Mobile phase B was 95% acetonitrile, 5% mobile phase A. The gradient was as follows at 150 μl/min flowrate: 0 min: 84.2 %B; 20 min: 21.1 %B; 22 min: 15.8 %B; 22.5 min: 84.2 %B; 30 min: 84.2 %B.

Data files were converted to mzML format using ProteoWizard msConvert (Chambers et al., 2012), and all peaks were manually reviewed using MAVEN2 software (Seitzer et al., 2022). Targeted compounds were matched by retention time and a secondary transition whenever possible. Each compound was fit to its own model. If data at both concentrations were available, it was fit to the model "$\log_2(abundance) \sim gene * concentration$." If only one concentration was available, it was fit to the model "$\log_2(abundance) \sim gene$." The P-values were then adjusted to $q$ values using the R package "q value" (Storey et al., 2025).

## Cytochrome *c* oxidase activity

Cytochrome *c* oxidase activity was determined using Cytochrome *c* Oxidase Assay Kit (Sigma-Aldrich) according to the manufacturer's instructions, modified to adapt the assay to a 96-well plate format. Isolated mitochondria (see "Mitochondrial isolation" above) were resuspended to 1 mg/ml in mitochondrial isolation buffer, then diluted 1:10 in enzyme dilution buffer with 1 mM n-dodecyl β-D-maltoside, and left on ice for 15 min before assaying. Reduced ferrocytochrome *c* substrate solution was prepared according to the manufacturer's instructions. Each condition and replicate were paired, with 10 μl mitochondrial suspension added to one signal well, and the equivalent volume of 1× enzyme dilution buffer added to the other background well. 95 μl 1× assay buffer (10 mM Tris-HCl, pH 7.0, containing the

appropriate concentration of KCl and NaCl) and 5 μl reduced ferrocytochrome *c* substrate solution were added to each well for a final volume of 110 μl. Absorbance at 550 nm was measured on a CLARIOstar Plus plate reader (BMG Labtech) immediately. The plate was read every 30 s for 20 cycles.

To calculate activity, the slopes describing absorbance at 550 nm vs time of each sample were determined using the "rlm" function from the R package "MASS" (Venables and Ripley 2002). For each experiment, the average slope of all the background conditions was subtracted from each sample and scaled to the amount of mitochondria added in milligrams to get the activity. Negative activities in the samples were set to 0. The model formula was "activity $\sim$ ionic_strength * buffer + experiment," where "ionic_strength" was treated as a factor.

## Internal potassium of young cells

Cell sampling and preparation were based on Eide et al. (2005). Strains were grown to saturation overnight in media containing the appropriate nitrogen source (ammonium sulfate or arginine) and standard (7.35 mM) potassium. From these overnight cultures, strains were diluted into media containing the appropriate potassium concentration in biological triplicate and grown for an additional 24 h, ensuring that the $OD_{600}$ did not go above ~0.1. The $OD_{600}$ of each culture was measured just prior to sampling. Samples of each media were also processed to estimate the amount of potassium collected by the filter. Enough volume to obtain a total of 1 $OD_{600}$ of cells was applied to a vacuum filtration system onto a 1.2-μm polycarbonate filter (RTTP02500, MilliporeSigma). This was washed 3 times with 5 ml of 1 μM EDTA disodium salt, followed by 3 washes of 5 ml water. Using forceps, the filter was placed into a screw-top microcentrifuge tube (HS10060, MilliporeSigma) containing 500 μl 35% $HNO_3$ such that it was completely submerged. During this process, ~1 ml of the culture was placed into a 50-ml Falcon tube and placed on ice for cell count and size analysis. The sample in $HNO_3$ was placed in a shaking incubator set to 65°C and left to digest overnight. Cell counts and cell diameter were determined on a CoulterCounter, as above. The previously collected samples on ice were diluted into 10 ml Isoton II Diluent (C96980; Beckman Coulter) such that ~5,000–10,000 cells were analyzed in a 500 μl sample.

The next day, each sample was processed by adding 500 μl water, vortexing, and removing the filter. The sample was centrifuged for 10 min at 12,000 *g*, and the supernatant was transferred to a 15-ml Falcon tube. To this, 7.75 ml water was added to achieve a final $HNO_3$ concentration of 2%.

Reference standards were prepared by pouring out 1 ml of a 1 g/l potassium ion reference solution (06335-100 Ml, MilliporeSigma) into a 1.5-ml tube. A 50-ml Falcon tube was filled with 50 ml blank solution (2% $HNO_3$). 200 μl of blank was removed and replaced with 200 μl of the potassium ion reference solution. This initial solution was diluted by serial transfer of 5 ml to four Falcon tubes each containing 20 ml blank solution. The final standard series contained 4,000, 800, 160, 32, and 6.4 ng/ml potassium ion.

Samples were run on a PerkinElmer 5300 DV optical emission ICP (OEICP) with an autosampler at UC Berkeley's Rausser

College of Natural Resources. Blank solution and reference standards were prepared fresh and run with every experiment. The OEICP signal was background-subtracted and converted into units of µg/l using the calibration curve generated from the reference standards. The amount of potassium collected on the filter in the blank samples was subtracted from all samples. Cell diameter was estimated as a function of genotype, potassium concentration, and nitrogen source using the model formula "mean_cell_diameter ~ gene + nitrogen + log(concentration)," and then used to calculate internal potassium concentration for each sample. Estimates for internal concentration were then analyzed using the formula "internal_concentration ~ genotype * external_concentration * nitrogen_source + (1|experiment)."

### Internal potassium of aged cells

Triplicate cell cultures were aged for 48 h on MADs as described above. Separately, triplicate cell cultures were inoculated such that they spent 24 h in the exponential phase at the end of the 48-h aging period. All cultures were harvested on the same day and processed as above (Internal potassium of young cells), with the following modifications. For the exponential cultures, 1 $OD_{600}$ was collected from each culture. For the 48-h cultures, the $OD_{600}$ of a 1:50 dilution of each culture was measured, and the total $OD_{600}$ corresponding to the most dilute culture was collected from each sample. Cell counts and volumes of all cultures were determined using the CoulterCounter, as above.

To avoid altering potassium ion content, the cultures were not fixed during the wash steps. This resulted in significant daughter cell contamination in the 48-h cultures (Fig. S5 D, orange, left peak), and required us to estimate the proportion and volume of old mothers in the 48-h culture. To do this, we modeled each size distribution as a mixture of three Gaussians using the "GaussianMixture" class from the Python package "scikit-learn" (Pedregosa et al., 2011), and designating the distribution with the smallest mean as "daughters." The two other distributions were summed to create an old cell size distribution. The potassium content of old cells was estimated in two ways. First (Fig. S5 E, purple), we assumed that the daughters of old mothers had the same potassium content as cells in exponential culture (Fig. S5 D, blue). Second, to get a maximum upper bound on the estimate (Fig. S5 E, brown), we made the (unlikely) assumption that the daughters of old cells contained no potassium (i.e., all of the extracted potassium came from old cells). In both cases, the concentration of potassium in old mothers was significantly lower than cells in an exponentially growing culture.

### Online supplemental material

Fig. S1 shows reporter values for individual cells as a function of different aging parameters. Fig. S2 shows control experiments characterizing the MMP reporter system. Fig. S3 shows data associated with the screening platform for finding mutants that affect the age-associated decline of MMP. Fig. S4 shows other phenotypes associated with the sis2Δ mutant. Fig. S5 shows additional control experiments and a model relating potassium ion homeostasis to mitochondrial function through the activity of

transporter Trk1p. Table S1 shows RNAseq data comparing young (exponentially growing) and 24-h old mother cells. Table S2 shows summary data for the deletion mutants and includes data from the initial screen, lifespan data, and the MMP ratio after 24 h compared with WT. Table S3 shows metabolomics data of the sis2Δ mutant compared with WT in standard (7.35 mM) KCl. Table S4 shows RNAseq data comparing WT and sis2Δ mutant in standard (7.35 mM) and low (0.5 mM) KCl. Table S5 shows a table of strains used in this study. Table S6 shows a table of oligos used to make the deletion mutants and check the deletions. Table S7 shows raw MMP and lifespan data of the remade deletion mutants and WT. Table S8 shows all other data used to generate the figures. Video 1 shows an example of a WT cell expressing the reporter system aging in the YLM, which is the same cell shown in Fig. 1 B.

### Data availability statement

The data underlying all figures are available in the published article and its online supplemental material. Raw RNAseq data underlying Table S1 and Table S4 are openly available in the NCBI Gene Expression Omnibus as GSE314946.

## Acknowledgments

The authors wish to acknowledge Tina Mahatdejkul-Meadows for her initial development of the MMP reporter system, Jonathan Paw for operating the FACS equipment during the screen sorts, and Wenbo Yang (UC Berkeley College of Natural Resources) for operating the OEICP machine.

Author contributions: Adam James Waite: conceptualization, data curation, formal analysis, investigation, methodology, project administration, software, supervision, validation, visualization, and writing—original draft, review, and editing. Beiduo Rao: data curation, investigation, methodology, validation, and writing—review and editing. Elizabeth Schinski: data curation and software. Nathaniel H. Thayer: conceptualization, data curation, formal analysis, methodology, resources, and software. Manuel Hotz: conceptualization, investigation, and writing—review and editing. Austin E. Y. T. Lefebvre: data curation, formal analysis, methodology, resources, software, validation, and writing—review and editing. Celeste Sandoval: data curation, methodology, validation, and writing—original draft. Daniel E. Gottschling: conceptualization, supervision, and writing—review and editing.

Disclosures: All authors have completed and submitted the ICMJE Form for Disclosure of Potential Conflicts of Interest. A.E. Lefebvre reported "Calico Life Sciences LLC provides me with employment." D. Gottschling reported "All authors are employees of Calico Life Sciences LLC. The work presented here is not of commercial value to Calico, but rather a contribution to advancing the study of cellular aging in yeast." No other disclosures were reported.

Submitted: 19 May 2025

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

**Supplemental material**

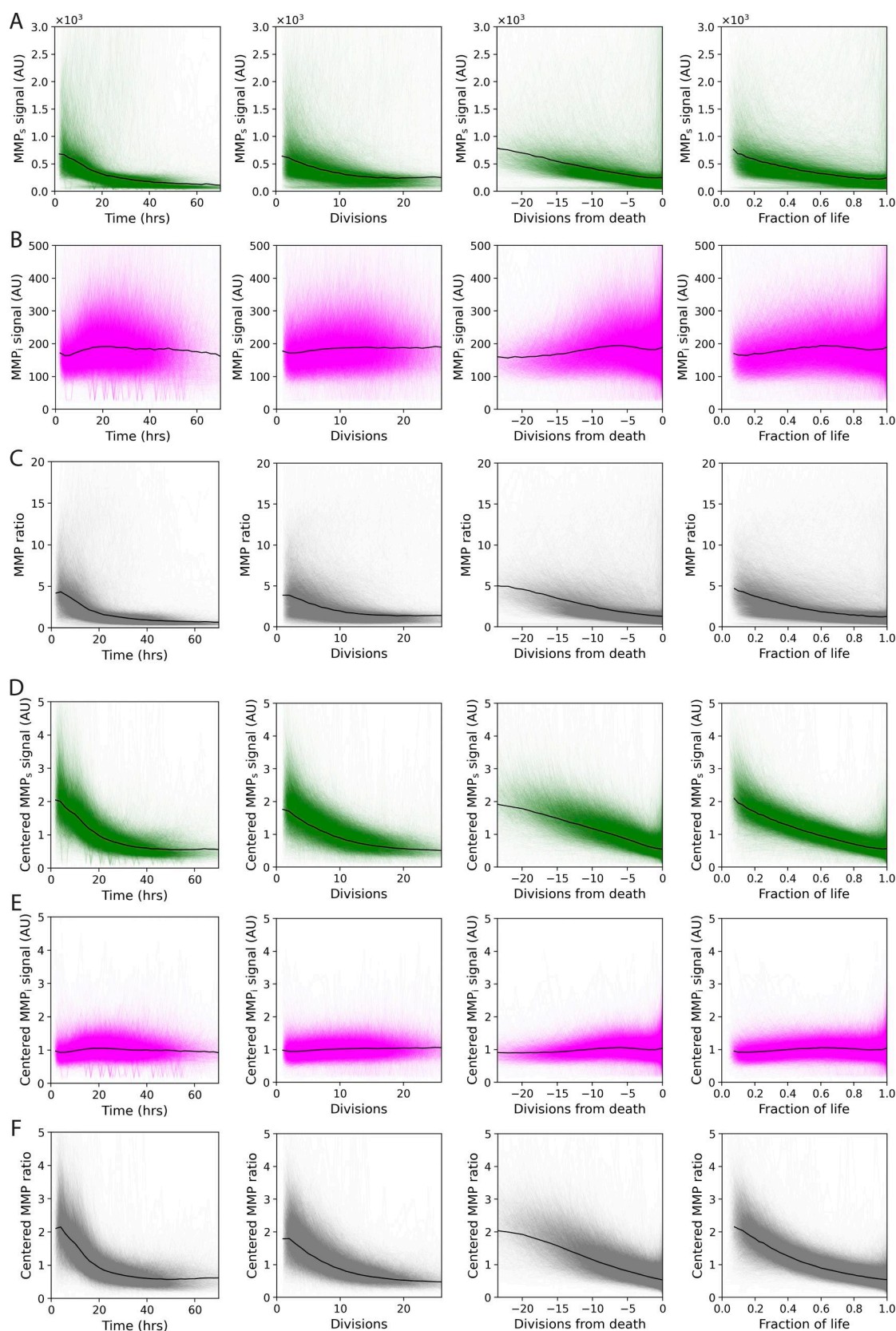

Figure S1. **Reporter values for individual cells as a function of different aging parameters. (A–C)** Raw data. Black line indicates median value. **(D–F)** Indicated reporter value of each cell was centered by dividing by its mean MMP ratio. Black line indicates the mean value. (A and D) MMP$_s$ (green), (B and E) MMP$_i$ (magenta), and (C and F) MMP ratio (MMP$_s$/MMP$_i$, gray). Values are shown from left to right as a function of time, number of divisions, number of divisions before death, and fraction of life as defined by the number of divisions. Data for divisions <1 were not included.

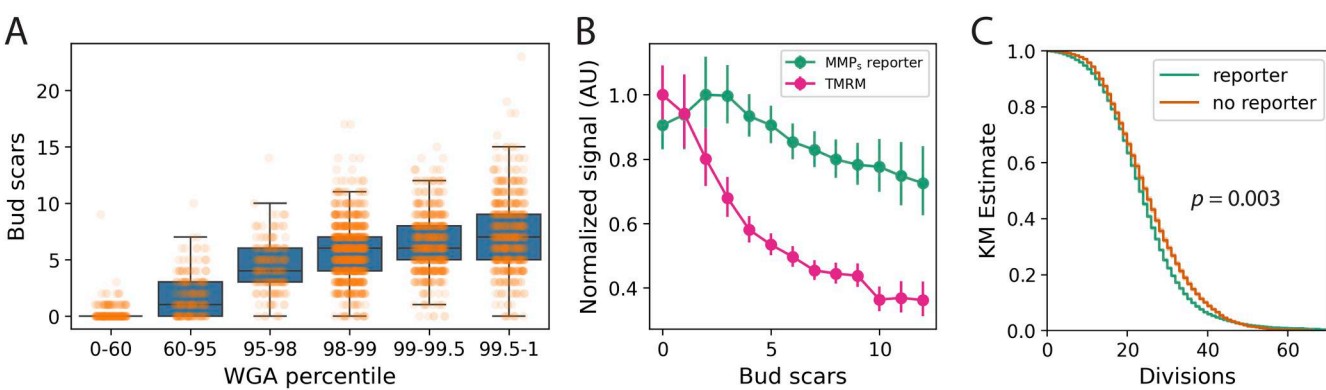

Figure S2. **Characterization of the MMP reporter system. (A)** Number of bud scars as a function of their WGA intensity (3 experiments, $n$ = 2,576 cells). Error bars show 1.5 times the interquartile range. **(B)** Intensity of the $MMP_s$ reporter component (green) and TMRM (magenta) measured in the same cells as a function of their age as determined by the number of bud scars. Data are normalized to the maximum value for each fluorescent marker so that the decline can be more easily compared. **(C)** Lifespans of strain with the full reporter system (green, 44 experiments, $n$ = 44,618 cells) and without it (orange, 11 experiments, $n$ = 17,099 cells). For B and C, error bars show 95% CI.

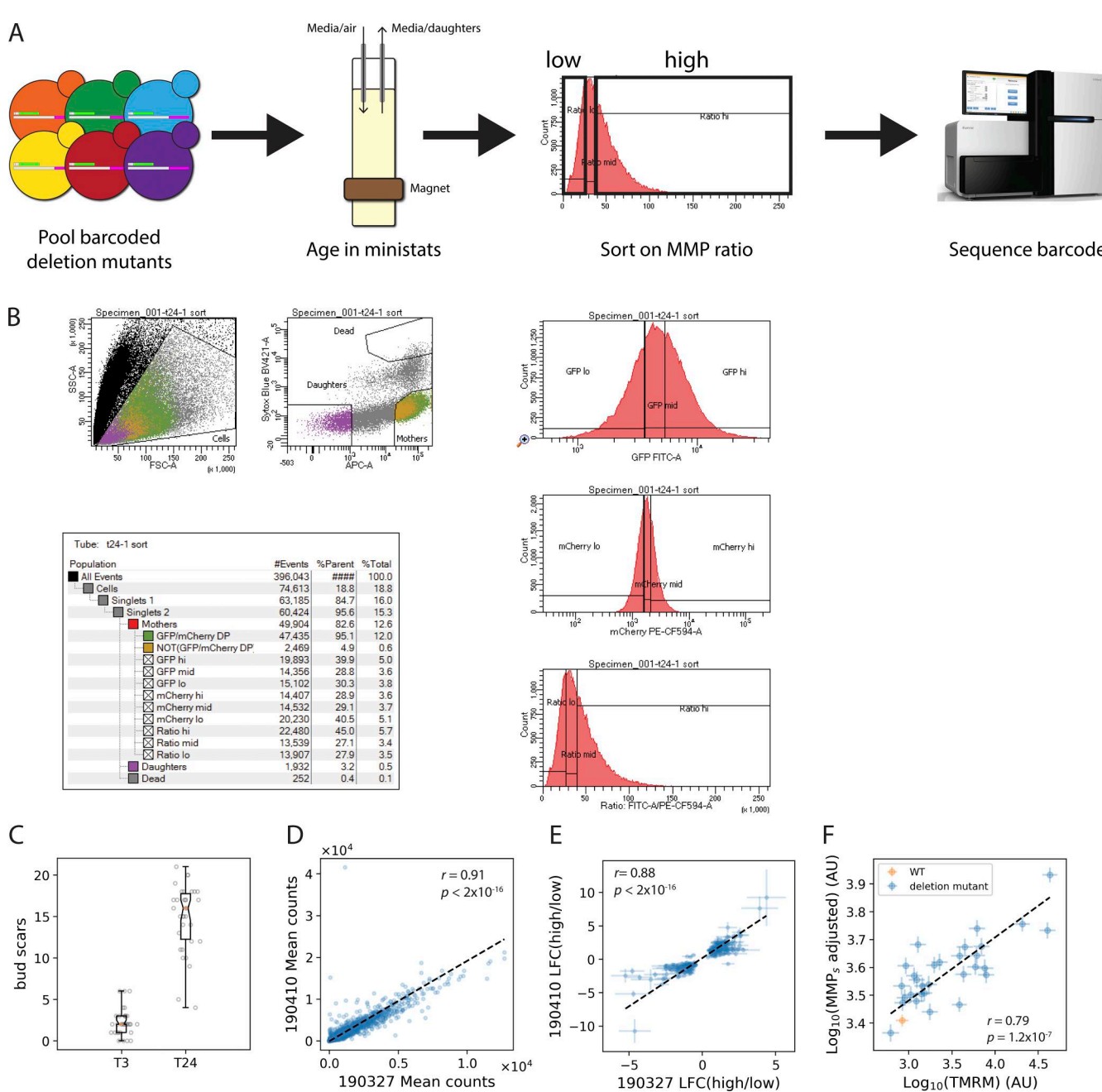

Figure S3. **Screening platform for finding mutants that affect the age-associated decline of MMP. (A)** Schematic of the screen. Image of sequencer courtesy of Illumina, Inc. **(B)** Raw data of the sort showing gating strategy. **(C)** After 3 (T3) and 24 (T24) hours, samples of the population were imaged on a fluorescent microscope and their bud scars were counted (T3, $n = 36$; T24, $n = 34$). **(D)** Correlation of barcode counts in two independent experiments ("190327" and "190410"). **(E)** Correlation between deletion mutants whose frequency significantly changed (adjusted P < 0.05) in the high vs low MMP ratio sorts. Error bars show 1 SEM. **(F)** Correlation between the $MMP_s$ portion of the fluorescent reporter, adjusted for mitochondrial mass (Materials and methods), and the MMP-sensitive dye TMRM in WT (orange) and each of the rescreened deletion mutants (blue). Error bars show 1 SEM. Plot shows data from six independent experiments. See Materials and methods for details. In **D–F**, the dashed line indicates best-fit line from Deming regression (Materials and methods). "$r$" indicates the Pearson correlation coefficient, and "$p$" is the P-value of $r$.

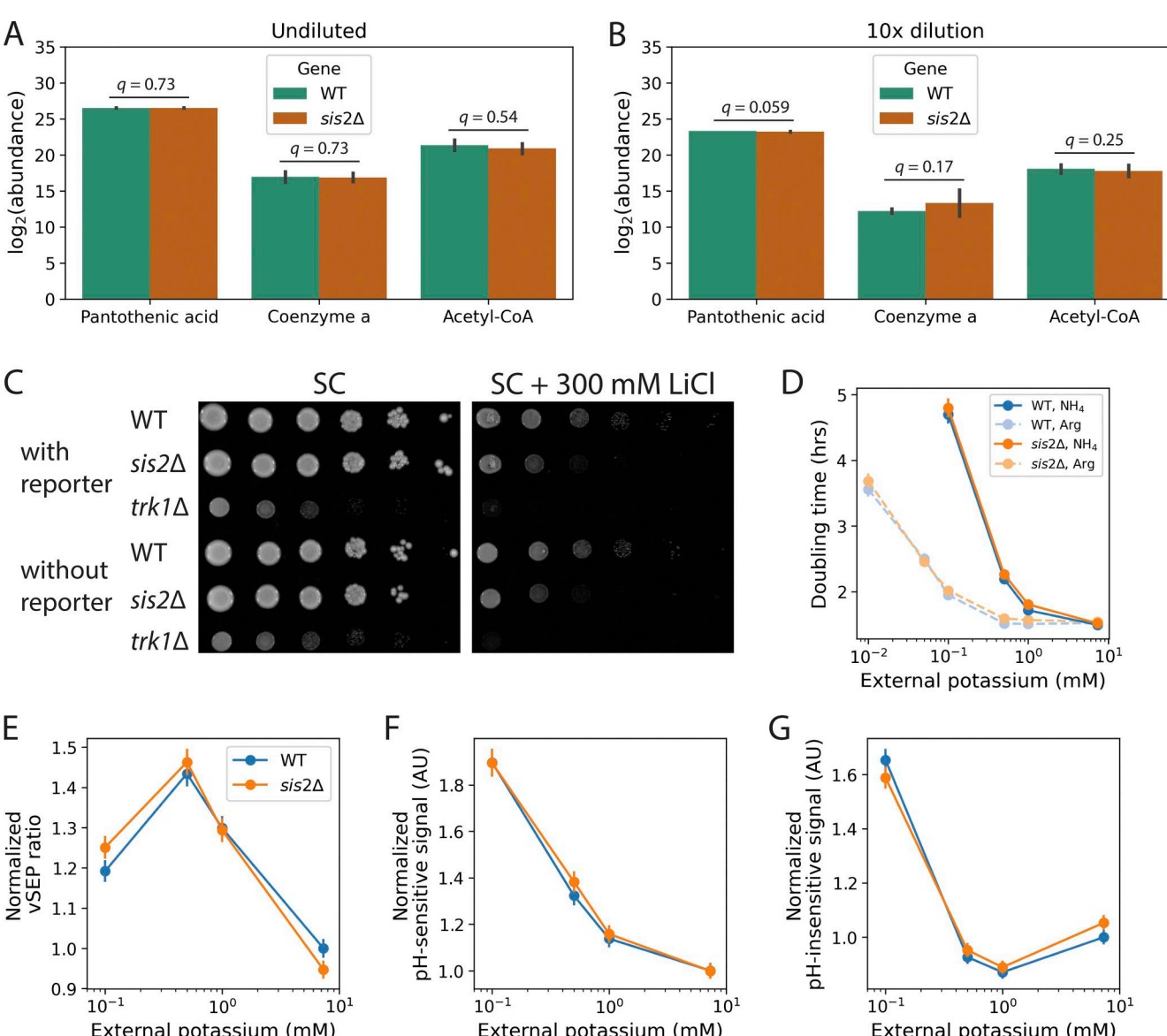

Figure S4. **Other phenotypes associated with the *sis2*Δ mutant. (A and B)** Abundance of several metabolites in the coenzyme A biosynthesis pathway as determined by targeted metabolomics on undiluted **(A)** or 10-fold diluted **(B)** extracts of strains without the reporter. Three biological replicates per strain. *q*-values are adjusted P-values (Materials and methods). Error bars show 2× SEM. **(C)** Growth tests on SC, pH 4.5 (left), or SC + 300 mM LiCl, pH 4.5 (right), for the indicated strains. Both plates are imaged after 2 days. **(D)** Culture doubling times of WT and *sis2*Δ in various environments. Data are from two experiments for each nitrogen condition. **(E)** vSEP ratio as a function of external potassium concentration. **(F)** pH-dependent and **(G)** pH-independent signals as a function of external potassium. Data are from two experiments. Biological triplicate measurements of two clones for each genotype were made, for a total of *n* = 6 replicates per genotype per experiment. Error bars show 95% confidence intervals.

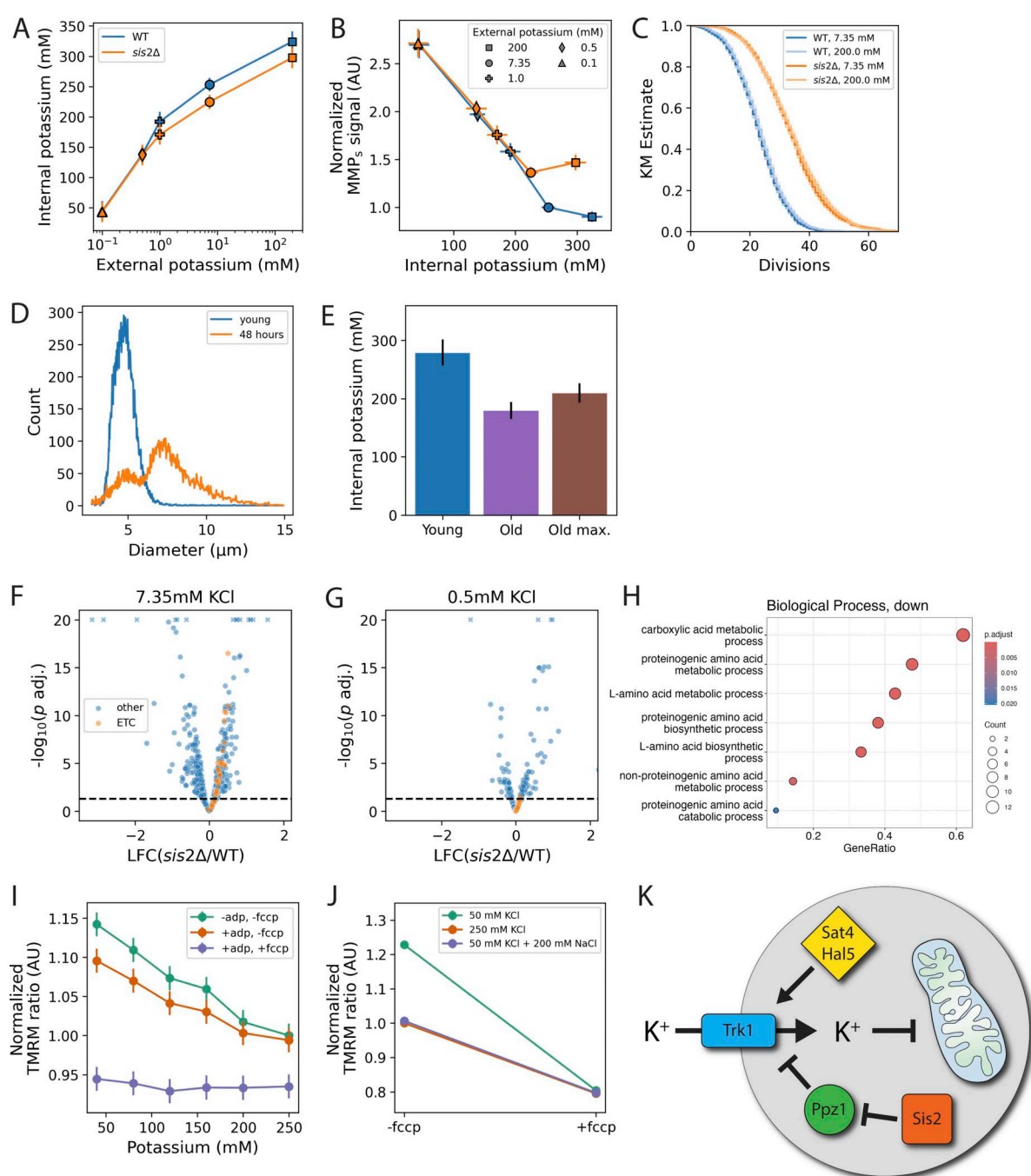

Figure S5. **Additional control experiments and a model relating potassium ion homeostasis to mitochondrial function through the activity of transporter Trk1p. (A)** Internal potassium as a function of external potassium including data for WT (blue) or *sis2*Δ mutants (orange) grown in ammonium and 200 mM KCl. **(B)** MMP$_s$ as a function of internal potassium for conditions shown in A. **(C)** Lifespan curves of strains grown in 7.35 mM (dark) or 200 mM (light) KCl. Data are from two experiments with $n$ = 3,214 for WT and $n$ = 2,789 for *sis2*Δ. **(D)** Cell diameter distributions of a representative young, exponentially growing culture (blue) and a culture that has been aged 48 h on the MAD (orange). **(E)** Concentration of potassium in the exponentially growing cultures (blue) and two estimates for old cells in the 48-h culture (right peak of orange curve). "Old" (purple) was calculated by assuming that daughters of old cells (left peak of orange curve in D) have same potassium content of cells from an exponential culture (blue curve in D). "Old max." (brown) provides the maximum upper bound on the potassium concentration of old cells, by making the (unlikely) assumption that daughters of old cells contain no potassium. Data are from two experiments, with three replicate "young" and old cultures per experiment. **(F and G)** RNAseq data from cultures grown in typical (7.35 mM, F) and low (0.5 mM, G) external potassium. Maximum -log$_{10}$($p$ adj.) was capped at 20. Truncated values are indicated by an "x." **(H)** Gene set enrichments (Materials and methods) for genes with less expression in *sis2*Δ vs WT in both potassium environments. **(I)** Mitochondria isolated from *sis2*Δ mutants were treated as in Fig. 7 A. Slopes of -adp, -fccp and +adp, -fccp conditions are significantly different from zero (P < 0.001 in each case). Differences between -adp, -fccp (green) and +adp, -fccp (orange) at KCl concentrations below 250 mM are significant with P ≤ 0.01. The difference between these groups at 250 mM is not significant (P = 0.54). **(J)** MMP (as measured by the TMRM ratio) of isolated mitochondria incubated in the indicated buffers. After measuring the TMRM ratio, FCCP was added to dissipate the MMP and samples were measured again. Data are from two experiments, with three replicates per condition. In all cases, error bars show 95% CIs. In **J**, error bars are smaller than the points. **(K)** Model depicting the relationship between potassium ion homeostasis and mitochondrial function.

Video 1.   Example cell expressing the reporter system and aging in the YLM, as imaged using epifluorescence microscopy (see Methods for details). Video shows image sets that included fluorescence, which were taken every 2 h (the bright-field channel was imaged every 15 min). The left panel shows bright-field; the right panel shows a composite of the preCOX4-mNeonGreen signal (green) and the TIM50-mCherry signal (magenta). These data were used to generate Fig. 1 B.

**Provided online are Table S1, Table S2, Table S3, Table S4, Table S5, Table S6, Table S7, and Table S8. Table S1 shows RNAseq data comparing young (exponentially growing) and 24-h old mother cells. Table S2 shows summary data for the deletion mutants and includes data from the initial screen, lifespan data, and the MMP ratio after 24 h compared with WT. Table S3 shows metabolomics data of the *sis2Δ* mutant compared with WT in standard (7.35 mM) KCl. Table S4 shows RNAseq data comparing WT and *sis2Δ* mutant in standard (7.35 mM) and low (0.5 mM) KCl. Table S5 shows a table of strains used in this study. Table S6 shows a table of oligos used to make the deletion mutants and check the deletions. Table S7 shows raw MMP and lifespan data of the remade deletion mutants and WT. Table S8 shows all other data used to generate the figures.**

