## [Peer Review File · The Journal of Cell Biology]

Potassium ion homeostasis modulates mitochondrial function

Adam Waite, Beiduo Rao, Elizabeth Schinski, Nathaniel Thayer, Manuel Hotz, Austin Lefebvre, Celeste Sandoval, and Daniel Gottschling

Corresponding Author(s): Daniel Gottschling, Calico Life Sciences LLC

Review Timeline:

Submission Date:	2025-05-19
Editorial Decision:	2025-06-25
Revision Received:	2025-11-11
Editorial Decision:	2025-12-03
Revision Received:	2025-12-10

Monitoring Editor: Thomas Langer

Scientific Editor: Gabriele Stephan

Transaction Report:

DOI: <https://doi.org/10.1083/jcb.202505110>

June 25, 2025

Re: JCB manuscript #202505110

Dan Gottschling
Calico Life Sciences LLC

Dear Dr. Gottschling,

Thank you for submitting your manuscript entitled "Potassium ion homeostasis regulates mitochondrial function". Your manuscript has been assessed by expert reviewers, whose comments are appended below. Although the reviewers express potential interest in this work, significant concerns unfortunately preclude publication of the current version of the manuscript in JCB.

You will see that the reviewers are overall supportive and think that your study presents interesting findings regarding potassium-based regulation of mitochondria during aging. The referees offer constructive feedback, and we think that your manuscript would improve if all of the reviewers' concerns could be addressed. This includes a more detailed analysis on the role of Sis2 in lifespan extension, measurements of cytochrome oxidase activity, as well as mechanisms underlying potential potassium changes in aged cells. We further agree with the reviewers and think that a more detailed characterization of the Cox4/Tim50-reporter should be included in a revised manuscript.

Please let us know if you are able to address the major issues outlined above and wish to submit a revised manuscript to JCB. We would like to ask you to provide us with a revision plan, outlining how you plan to address the reviewers' critiques prior to resubmission. Note that a substantial amount of additional experimental data likely would be needed to satisfactorily address the concerns of the reviewers. The typical timeframe for revisions is three to four months. If you anticipate any difficulties in meeting this aforementioned revision time limit, please contact us and we can work with you to find an appropriate time frame for resubmission. Please note that papers are generally considered through only one revision cycle, so any revised manuscript will likely be either accepted or rejected.

If you choose to revise and resubmit your manuscript, please also attend to the following editorial points. Please direct any editorial questions to the journal office.

GENERAL GUIDELINES:

Text limits: Character count is < 40,000, not including spaces. Count includes title page, abstract, introduction, results, discussion, and acknowledgments. Count does not include materials and methods, figure legends, references, tables, or supplemental legends.

Figures: Your manuscript may have up to 10 main text figures. To avoid delays in production, figures must be prepared according to the policies outlined in our Instructions to Authors, under Data Presentation, <https://jcb.rupress.org/site/misc/ifora.xhtml>. All figures in accepted manuscripts will be screened prior to publication.

*****IMPORTANT:** It is JCB policy that if requested, original data images must be made available. Failure to provide original images upon request will result in unavoidable delays in publication. Please ensure that you have access to all original microscopy and blot data images before submitting your revision. *******

Supplemental information: There are strict limits on the allowable amount of supplemental data. Your manuscript may have up to 5 supplemental figures. Up to 10 supplemental videos or flash animations are allowed. A summary of all supplemental material should appear at the end of the Materials and methods section.

Please note that JCB now requires authors to submit Source Data used to generate figures containing gels and Western blots with all revised manuscripts. This Source Data consists of fully uncropped and unprocessed images for each gel/blot displayed in the main and supplemental figures. For assays performed using capillary electrophoresis and/or immunoassay-based detection, authors should instead provide the electropherogram graph(s) for each experiment, plotting fluorescence/chemiluminescence intensity vs. molecular weight/size. Please be sure to provide one Source Data file for each figure gels, blots, and/or capillary electrophoresis assays along with your revised manuscript files. File names for Source Data figures should be alphanumeric without any spaces or special characters (i.e., SourceDataF#, where F# refers to the associated main figure number or SourceDataFS# for those associated with Supplementary figures). For traditional gels and blots, the lanes of the gels/blots should be labeled as they are in the associated figure, the place where cropping was applied should be marked (with a box), and molecular weight/size standards should be labeled wherever possible. For capillary electrophoresis assays, each trace in the graph should be color-coded and labeled to indicate which protein, gene, or sample is being measured (please try to avoid red/green combinations to accommodate our color-blind readers).

If you choose to resubmit, please include a cover letter addressing the reviewers' comments point by point. Please also highlight all changes in the text of the manuscript.

Regardless of how you choose to proceed, we hope that the comments below will prove constructive as your work progresses. We would be happy to discuss them further once you've had a chance to consider the points raised. You can contact the journal office with any questions at cellbio@rockefeller.edu.

Thank you for thinking of JCB as an appropriate place to publish your work.

Sincerely,

Thomas Langer
Monitoring Editor
Journal of Cell Biology

Gabriele Stephan
Scientific Editor
Journal of Cell Biology

Reviewer #1:

Waite et al:

Potassium ion homeostasis regulates mitochondrial function

In the current manuscript, the authors address the role of changes in the mitochondrial membrane potential as a factor in cellular aging. To address this fundamental question, they designed a new genetically encoded sensor/reporter system for mitochondrial membrane potential (MMP) in budding yeast during replicative aging. Using this reporter, the authors performed high-throughput analysis of MMP in replicatively aging cells and confirm a decrease of MMP with age. Based on the reporter system, cells displayed an initial rapid decline in MMP within the first 10 divisions followed by a second slower decline until the end of the lifespan. Interestingly, the average MMP during the first 5 divisions significantly correlated with replicative lifespan suggesting increased MMP is a predictor of longer lifespan. Next, the authors conducted a genome-wide screen to identify mutants with increased mitochondrial membrane potential (MMP) during replicative aging. One of the hits, the gene deletion of *SIS2*, previously linked to longevity, caused increased MMP and displayed the largest extension of replicative lifespan. The authors explored the genetic interactions of *sis2* and known factors, which suggested a link to intracellular potassium homeostasis for MMP *in vivo* and *in vitro*.

This study combines new tool development, unbiased genome-wide genetic screening, and genetic analysis with *in vivo* and *in vitro* approaches. It provides compelling evidence that ion homeostasis constitutes a critical factor for MMP, mitochondrial function and cellular aging. Although several links (*sis2*-induced longevity, *sis2* connections to *Trk1* regulation, potassium homeostasis effects on longevity) have been shown individually before (and all studies are cited by the authors), this study combines these individual observations into a coherent model centering on MMP regulation during cellular aging. Nevertheless, there are a few critical points that need to be addressed as outlined below.

Major points:

(1) The Cox4/Tim50-reporter system for MMP requires additional characterization.

The authors present a novel genetically encoded reporter system to assess MMP. However, the manuscript lacks a thorough and comparative analysis of this new reporter.

The authors should clarify the logic of their reporter system. As they state, both, preCox4-mediated and Tim50 import into mitochondria is MMP-dependent. Thus, the difference in sensitivity towards changes in MMP is significantly based on their differential protein stability/dilution during cell division rather than MMP. However, the authors do not provide the experimental data to compare the behavior of both reporters over time during aging. Both mNeonGreen (MMPd) and Tim50 (MMPi) signals need to be shown in figure S1. Why is mNeonGreen less stable or more diluted in mitochondria in dividing mother cells than Tim50? Additionally, how does Tim50, which is MMP-dependent, compare to an outer mitochondrial protein like Tom70 or others, which is truly independent of MMP for insertion during aging? The claim in line 86 is not supported at this point. It should say "is designed to report on MMP-dependent import capacity". The authors would need to provide experimental evidence for their claim. Importantly, how does the preCOX4-SL17-mNeonGreen/Tim50-mCherry reporter compare to established MMP-

dependent dyes like DiOC6 or TMRM in detecting age-associated changes in MMP in terms of sensitivity and kinetics? Given the biphasic behavior of MMP during aging shown in Figure 1C, this is important to characterize.

The authors should test the reporter system under different carbon sources (glucose, galactose, glycerol). Because Tim50 is under endogenous control but preCOX4-SL17-mNeonGreen is under the control of the TPI1 promoter, does this system report on lower MMP in the presence of galactose or glycerol, which derepress or induce mitochondrial biogenesis and Tim50 levels but not preCOX4-SL17-mNeonGreen? This is important to analyze to understand the limitations of the reporter itself and of the genetic screen described in the manuscript.

Does the presence of the reporter system affect cellular aging? (Over)expression of matrix-localized fluorescence reporters may cause a significant burden on protein import in mitochondria with potential effects on replicative aging. Thus, the authors need to test replicative aging of cells expressing the reporter system or not.

The efficacy of the SL17 degron is not shown and should be included in the supplementary data.

Why do the authors use MMPd instead of the ratio in Figure S2F? Here the authors might conflate differences in MMP with mitochondrial mass (as seen in figures 5C and D).

(2) Mechanistic analysis of *sis2* lifespan extension

The mechanism, through which *sis2* deletions elevate MMP and extend lifespan is inferred from genetic analyses. It would be important to confirm the key hypotheses of the study by additional orthogonal approaches. For example, the authors propose that *sis2* functions through Ppz1 and the regulation Trk1. However, changes in Trk1 protein levels or activity are not directly tested. This is particularly important, because, as the authors discuss, reducing external potassium still increased the lifespan of *sis2*, suggesting additional lifespan-extending effects. Specifically, the authors observe that in WT cells, replicative lifespan increases with elevated MMP (Figure 6C). However, *sis2* mutants seem to be only mildly affected by further increased MMP upon decreased intracellular potassium concentrations, suggesting SIS2 deletion significantly affects replicative lifespan independent/in addition to MMP or there is a limited by which MMP can increase lifespan. A key question is: is elevated MMP required for the longevity phenotype of *sis2* mutants. For example, do increased potassium concentrations reduce the MMP and lifespan of *sis2* mutants? Does *sis2*-induced longevity require mitochondrial respiration to drive MMP?

The authors propose that *sis2* affects membrane potential via potassium ion homeostasis. To exclude effects on mitochondrial protein composition, the authors should check the steady state levels of RC proteins and complex assembly. Transcriptomics analysis of *sis2* mutants by Olmez et al suggest upregulation of oxidative phosphorylation genes.

The authors propose that deletion of SIS2 results in increased Ppz1 activity. Is PPZ1 overexpression sufficient to increase MMP and lifespan?

Given that *sat4* and *hal5* effects on lifespan and MMP are additive to *sis2*, suggests they are not functionally linked. Rather, a common function would suggest an epistatic relationship, which the authors seem to not observe.

Minor points:

Figure S2F: there are multiple datapoints labeled as "WT". What do they represent?

Figure 4A and B need include the *sis2trk1* double mutant.

Does intracellular potassium change with cellular age? Cell volume increases with age. Does this affect the intracellular potassium concentration?

Does intracellular potassium affect vacuolar acidity during aging? The authors test the effects on the vacuole in young but not in aged cells (Figure S4). This is an important control.

Reviewer #2:

The loss of mitochondrial activity is a hallmark of aging, but the underlying mechanisms remain unclear. In the present study, the authors used a sophisticated screening system to monitor the decline in mitochondrial protein import efficiency in budding yeast during aging. They compare the ratios of two mitochondrial fusion proteins, preCox4-degron-NeonGreen and Tim50-mCherry and observed a relative loss of the NeonGreen signal in older mother cells. Next, they screened for deletion mutants with higher green-over-red signals, thereby identifying a SIS2 deletion mutant as extreme case. A central role of Sis2 as regulator for yeast replicative lifespan was already shown by the Hochstrasser lab before (Olmez et al, Nat Comm, 2023). The authors further observed that the increased mitochondrial membrane potential in aging cells is connected to changes in potassium levels, even though the data of this second part of the study are not entirely conclusive. The impact of potassium on mitochondrial activity is mainly based on two conclusions: First, the potassium concentration in the medium is relevant for the measured green-over-red signals and very low as well as very high potassium concentrations correlate with less mitochondrial activity. Furthermore, the potassium concentration of the buffer affects the membrane potential of isolated mitochondria to some degree. Both observations are no strong evidence that in vivo the intracellular potassium concentration plays a direct role for the age-dependent decline of mitochondrial activity.

Thus, the study uses an elegant system to systematically screen for mutants that affect the accumulation of specific mitochondrial proteins. The Gottschling lab is world-leading for such screens and the technology here is impressive. Unfortunately, from the specific screen shown here, not so much can be learned. The authors demonstrate that Sis2 is a critical factor in respect to longevity and respective mutants show delayed aging symptoms; this is not novel. The underlying mechanism remains vague and the links between potassium, membrane potential, protein import and aging are not clear. Therefore, I feel that at least in its present form the study is only of limited interest for a broader readership.

Specific points:

1. The title claims that potassium ion homeostasis regulates mitochondrial function. First of all, the authors did not show that potassium regulates, i.e. controls, mitochondrial activity. They just show that the potassium concentration in the medium or in the buffer influences respiration. This would presumably also be the case if the authors would have altered the levels of magnesium, sodium or ammonium.
2. The green-over-red screen to visualize Cox4-over-Tim50 ratios is interesting as it suggests that mitochondria alter the protein import efficiency on a protein specific level. However, the authors just use this interesting aspect as a proxy for membrane potential which they could have analyzed much easier with a membrane potential-sensitive dye. It is not clear whether the lower import of Cox4 is due to a reduced membrane potential or vice versa, the reduction in the import of OXPHOS enzymes such as Cox4 reduces the membrane potential. This initial part of the study is very superficial. A more comprehensive analysis why or how the import of some proteins changes during aging would have made this study much more interesting.
3. It is not entirely clear how the green-over-red screen works. The authors refer to Martin Ralser's study who however used different fusion proteins. The changes in signal intensities could be due to changes in protein stability rather than membrane potential-dependent import. It is essential to carefully analyze what this reporter exactly measures.
4. The authors need to exclude that the decline of the green-over-red signal is caused by a different bleaching behavior of the two proteins. They might simply use dead cells for which the ratio would have to stay constant over time.
5. In the supplemental table, the authors provide the specific values of their fluorescence measurements in a test set of different mutants. The table lacks a description, but from what I can see, the total fluorescence intensities in the mutants are extremely heterogeneous. For example, a deletion of DEG1 shows a 10 fold higher log fold change of the GFP signal than the SIS2 mutant that was further analyzed in the study. Does this mean that the Cox4 fusion protein is accumulating in the absence of DEG1 to about 1,000 fold higher levels than in the SIS2 deletion mutant? How can a reporter be reliable if the expression intensities differ so much? Even ratiometric sensors are only reliable if the expression intensities are comparable.
6. The authors used glucose as carbon source for their experiments. However, many mitochondrial proteins are glucose-repressed. The screen did not bring up any mutant that lacks a mitochondrial protein but many mutants that affect metabolism. Doesn't this suggest that the screen does not report about the age-dependent import behavior of Cox4 and Tim50, but rather about changes in the metabolism of aging cells which affects mitochondrial function only indirectly?
7. The authors propose that potassium influences cytochrome oxidase activity and that this is the basis of the age-dependent decline in mitochondrial function. It should be easy to measure cytochrome oxidase activity or levels in young vs old cells in wild type and TRK1 mutants.

Reviewer #3:

In this manuscript, Waite et al. report the role of potassium ions in cellular longevity through affecting mitochondrial membrane potential (MMP) in yeast. Previous studies reveal that mitochondrial dysfunction is one of the major factors associated with aging and aging-related disorders in many cell types. Although numerous genome-wide and systematic screens have so far identified genes and proteins whose loss leads to reduced or increased lifespan, cellular pathways whose defects affect MMP and lifespan have not yet been comprehensively explored. In this study, the authors established a reporter system to monitor MMP in the yeast *Saccharomyces cerevisiae* and found that MMP indeed declines during aging. Using ~4,700 single gene deletion strains expressing the MMP reporter, they performed a genome-wide screen for mutants exhibiting suppression of age-associated MMP decline and identified 21 candidates with high MMP and longer lifespan. Those mutants include the longest-lived one lacking *Sis2* whose mitochondrial oxygen consumption rate is elevated, possibly due to hyperactive mitochondria. Additional data suggest that *Sis2*, a negative regulatory subunit of the protein phosphatase *Ppz1*, activates *Trk1*, a plasma membrane potassium transporter, in a manner independent of *Sat4* and *Hal5*, two protein kinases responsible for *Trk1* activation. Importantly, loss of *Trk1* or potassium limitation led to an increase in MMP and lifespan. Furthermore, MMP of isolated mitochondria declined with increased potassium ions in vitro. Collectively, these findings imply that potassium ion homeostasis is tightly linked to MMP, which directly affects mitochondrial fitness, ultimately leading to alterations in lifespan.

The data in this manuscript are convincing with proper controls and significant statistics to support the conclusion that the major monovalent cation is one of the key aging-related factors acting through its direct negative action on mitochondrial function. Although it remains uncertain how much intramitochondrial potassium ion levels fluctuate under physiological conditions that could change mitochondrial activity, and how potassium ions affect the OXPHOS complexes, the findings in this study could provide new insights into the molecular mechanisms underlying cytoplasmic potassium ion-mediated regulation of mitochondrial function. Still, the authors need to address the following points and strengthen this manuscript.

Specific points:

1. The authors should investigate whether the cytoplasmic potassium ion levels are high in aged wild-type cells. If that is the case, which factor(s) could be the primary cause of alterations in potassium ion levels?
2. It would be interesting to see if loss of *Sis2* can ameliorate short lifespan phenotypes of mutants that are unrelated to

potassium ion homeostasis.

3. The authors should try to test if the OXPHOS complex IV activity is indeed increased in cells lacking Sis2.

Dear Gabriele and Thomas,

We very much appreciated the constructive feedback from the reviewers and you on our manuscript **Potassium ion homeostasis regulates mitochondrial function**. We now return with our revised manuscript that has been improved by carrying out a number of experiments, re-analyzing data and editing that helps to clarify points brought up in the reviews. We address all the reviewers' concerns in detail below. Importantly, we provide additional information about the role of Sis2 in lifespan extension, have measured cytochrome oxidase activity, and examined potassium changes in aged cells. We also provided a more detailed characterization of the Cox4/Tim50-reporter.

Thank you for reconsidering our manuscript and I look forward to hearing from you soon,

dan

Reviewer #1

Major points

The Cox4/Tim50-reporter system for MMP requires additional characterization.

The authors present a novel genetically encoded reporter system to assess MMP. However, the manuscript lacks a thorough and comparative analysis of this new reporter.

Thank you for the feedback about the reporter system we developed and employed. We have taken your recommendations to heart and have carried out some additional experiments as well as written a more detailed explanation of the system along with justifications for our choices. In particular, we rewrote the section “Mitochondrial membrane potential of young cells is correlated with lifespan” (lines 77-134); we added lines 648-651 to the Methods; we added a new Fig. S1 dedicated to characterizing the reporter system; and we added more panels to Fig. S2 to show the trajectories of the different components of the reporter system. We address the comments more specifically below.

The authors should clarify the logic of their reporter system. As they state, both, preCox4-mediated and Tim50 import into mitochondria is MMP-dependent. Thus, the difference in sensitivity towards changes in MMP is significantly based on their differential protein stability/dilution during cell division rather than MMP. However, the authors do not

provide the experimental data to compare the behavior of both reporters over time during aging. Both mNeonGreen (MMPd) and Tim50 (MMPi) signals need to be shown in figure S1.

We have clarified the reasoning behind our use of TIM50 (lines 98-106) and have added separate figures for preCOX4-mNeonGreen, Tim50-mCherry, and the MMP ratio to Fig. S2 (which used to be Fig. S1). Our reasons for choosing Tim50-mCherry are based on things we learned from earlier published results (Hughes and Gottschling, 2012; Hughes *et al.*, 2016). Specifically, that Tim50-mCherry was very representative of mitochondria, did not enter into the age-associated Mitochondrial Derived Compartments (MDCs – which are destined for vacuoles and only contain Tom70 and outer membrane proteins), and, as we reinforce in Figure S2, that Tim50-mCherry remains at a virtually constant level across the entire lifespan of a WT mother cell. All together these data support our reason for choosing it as the “denominator” in the MMP ratio. We have also changed the descriptions of the two reporters from “MMP-dependent” and “MMP-independent” to “MMP-sensitive” and “MMP-insensitive” throughout the paper, which we believe more accurately reflects the nature of these reporters.

Why is mNeonGreen less stable or more diluted in mitochondria in dividing mother cells than Tim50?

While we have not tested this directly, we believe that mNeonGreen is imported into the mitochondrial matrix, where it can be turned over by the various matrix proteases, while Tim50 is an essential, stable member of the mitochondrial inner membrane, which is found in cells that lack MMP (Hughes and Gottschling, 2012; Hughes *et al.*, 2016).

Additionally, how does Tim50, which is MMP-dependent, compare to an outer mitochondrial protein like Tom70 or others, which is truly independent of MMP for insertion during aging?

See above.

The claim in line 86 is not supported at this point. It should say "is designed to report on MMP-dependent import capacity". The authors would need to provide experimental evidence for their claim.

Thanks for your comment here that led us to describe the reporter system with greater detail and clarity.

Importantly, how does the preCOX4-SL17-mNeonGreen/Tim50-mCherry reporter compare to established MMP-dependent dyes like DiOC6 or TMRM in detecting age-associated changes in MMP in terms of sensitivity and kinetics? Given the biphasic behavior of MMP during aging shown in Figure 1C, this is important to characterize.

To address this concern, we used a microscopy-based assay to observe the MMP_s (preCOX4-mNeon) portion of the reporter with TMRM in the same cells that were of different ages (Fig. S1A,B). These experiments showed that both measurements of MMP decline with age, but the mNeonGreen signal declines more slowly – which likely reflects the turnover rate of mNeonGreen. Thus, our reporter is a conservative estimate of MMP. We have added these results in Fig. S1 and have removed references to the biphasic nature of the decline. Understanding the underlying biology of this biphasic phenomenon is difficult to test and not a critical aspect of our overarching hypothesis that the reporter serves as a means to evaluate MMP in aging and can be employed for screening purposes.

The authors should test the reporter system under different carbon sources (glucose, galactose, glycerol). Because Tim50 is under endogenous control but preCOX4-SL17-mNeonGreen is under the control of the TPI1 promoter, does this system report on lower MMP in the presence of galactose or glycerol, which derepress or induce mitochondrial biogenesis and Tim50 levels but not preCOX4-SL17-mNeonGreen? This is important to analyze to understand the limitations of the reporter itself and of the genetic screen described in the manuscript.

To directly address the main concern, we compared RNA transcripts of young cultures with mother cells aged for 24 hours on the Ministat Aging Devices (MAD) and found that there is no difference in the expression of the TPI1 promoter or the TIM50 promoter under the conditions we performed our screening and testing in. We have added these results to the Methods section (lines 648-651). We agree that, if we were to use this reporter system with other carbon sources, we would have to recharacterize the reporter in these new carbon sources.

Does the presence of the reporter system affect cellular aging? (Over)expression of matrix-localized fluorescence reporters may cause a significant burden on protein import in mitochondria with potential effects on replicative aging. Thus, the authors need to test replicative aging of cells expressing the reporter system or not.

In designing the reporter system described here we were cognizant of potentially “overloading” the mitochondrial protein import system with the reporter proteins. Therefore, we tested a number of different promoters and reporter constructs before settling on the ones employed. In our assessment, we compared the lifespans of ~45,000 cells across 44 experiments with the reporter to ~17,000 cells across 11 experiments without the reporter and found that the strain with the reporter lived on average 1.4 divisions (~5%) shorter than the strain without the reporter. This difference seemed acceptable, considering it is less than the difference typically considered to be “significant” in the literature. We added this information to Fig. S1 and lines 648-651 in the Methods section.

In addition, and already included in the text, cells with initially higher MMP (i.e., *more* import of the reporter construct) on average live *longer* than cells with lower MMP (i.e., *less* import of the reporter); see Fig. 1E. If localization of the preCOX4-mNG reporter to the matrix had a

significant negative effect on replicative lifespan, we would expect that higher MMP would be associated with a shorter lifespan.

The efficacy of the SL17 degron is not shown and should be included in the supplementary data.

We have added to the text (lines 88-96) a more detailed description of the previously characterized efficacy of SL17 as a degron on different tagged proteins (Gilon, Chomsky and Kulka, 1998; Shlevin *et al.*, 2007; Papić *et al.*, 2013), and how mitochondrial matrix proteins tagged with this degron are protected from degradation (Shlevin *et al.*, 2007; Papić *et al.*, 2013). Consistent with this, in Fig. 1B of our paper the preCOX4-SL17-mNG reporter is concentrated in the mitochondria and does not accumulate in the cytoplasm even though the preCOX4-mNG signal has declined. This notably contrasts with Fig. 1B of (Vowinckel *et al.*, 2015), where a preCOX4-mCherry reporter (lacking a degron) shows dramatic cytoplasmic accumulation of mCherry.

Why do the authors use MMP_d instead of the ratio in Figure S2F? Here the authors might conflate differences in MMP with mitochondrial mass (as seen in figures 5C and D).

Originally, we compared MMP_d (now MMP_s) to TMRM since TMRM is not ratiometric and cannot be adjusted to account for mitochondrial mass (we now explain this in more detail in lines 81-86). To address the reviewer's concern, we have adjusted the MMP_s values to account for differences in the MMP_i signal (i.e., mitochondrial mass) using data we have from the YLM devices. This process is described in the Methods section, lines 886-896. In addition, to simplify the result, we are simply presenting the top 35 mutants identified in the screen and presented in Table 2 (see paragraph starting on line 148 for further explanation). Hopefully, it is clear that this still gives a very strong correlation.

Mechanistic analysis of *sis2* lifespan extension.

*The mechanism, through which *sis2* deletions elevate MMP and extend lifespan is inferred from genetic analyses. It would be important to confirm the key hypotheses of the study by additional orthogonal approaches. For example, the authors propose that *sis2* functions through *Ppz1* and the regulation *Trk1*. However, changes in *Trk1* protein levels or activity are not directly tested. This is particularly important, because, as the authors discuss, reducing external potassium still increased the lifespan of *sis2*, suggesting additional lifespan-extending effects.*

Trk1 activity appears to be independent of transcription- or translation-level changes (Yenush, 2016). Consistent with this, we did not see changes between WT and *sis2*Δ in transcript expression of *TRK1* by RNAseq, changes in protein levels of *Trk1p* by proteomics, or changes in expression of N- or C-terminally tagged *Trk1p*. We directly address the concern about the potassium-independent component of the longer lifespan of *sis2*Δ mutants below.

Specifically, the authors observe that in WT cells, replicative lifespan increases with elevated MMP (Figure 6C). However, sis2 mutants seem to be only mildly affected by further increased MMP upon decreased intracellular potassium concentrations, suggesting SIS2 deletion significantly affects replicative lifespan independent/in addition to MMP or there is a limited by which MMP can increase lifespan.

We agree that the *sis2*Δ mutant must have other lifespan-extending effects, since it has a longer lifespan than WT in both nitrogen sources and all external potassium concentrations that are not impacted by ammonium toxicity (Fig. 5B), and we are careful to mention this in the manuscript.

To come up with some hypothesis about differences between WT and *sis2*Δ, regardless of the potassium environment, we performed RNAseq on these two strains in “regular” (7.35 mM) and limited (0.5 mM) potassium, and found that expression of amino acid biosynthesis genes are lower in *sis2*Δ relative to WT (Fig. S8). Consistent with this, our metabolomics data revealed that amino acids and amino acid metabolic intermediates were significantly lower in the *sis2*Δ strain (Table S3). These findings reveal an area for future exploration of *Sis2*’s additional contributions in determining lifespan.

A key question is: is elevated MMP required for the longevity phenotype of sis2 mutants. For example, do increased potassium concentrations reduce the MMP and lifespan of sis2 mutants?

This is an interesting question, but we do not have a way to disrupt the MMP within *sis2*Δ cells in a way that does not also affect lifespan. Nevertheless, we increased external potassium to 200 mM and found the lifespans and MMP of both strains were unaffected (Fig. S5A-C). Thus, at high potassium it seems that some other homeostatic processes, such as vacuolar sequestration (Herrera *et al.*, 2013) or alternative feedback mechanisms, are protecting the cell from having too much potassium. We added these results to the manuscript (lines 358-361).

Does sis2-induce longevity require mitochondrial respiration to drive MMP?

While this is an interesting question, we believe answering it would be quite difficult. In principle, one way to address it would be to eliminate mitochondrial DNA, which would abolish respiration, and then create a mitochondrial membrane potential by expressing the *ATP1-111* allele that is able to generate a membrane potential in the absence of mitochondrial DNA (Veatch *et al.*, 2009). However, in this earlier work, we discovered that eliminating mitochondrial respiration in growing culture leads to a “crisis” that results in a hyper-mutable state (Veatch *et al.*, 2009). The resulting compensatory mutations – which could extend or shorten lifespan – would preclude an interpretable conclusion.

The authors propose that sis2 affects membrane potential via potassium ion homeostasis. To exclude effects on mitochondrial protein composition, the authors should check the steady state levels of RC proteins and complex assembly. Transcriptomics analysis of sis2 mutants by Olmez et al suggest upregulation of oxidative phosphorylation genes.

The RNAseq experiment mentioned above revealed that, as shown elsewhere, *sis2Δ* mutants have upregulated ETC transcripts (Fig. S5F). This difference goes away when WT and *sis2Δ* are grown in low (0.5 mM) potassium (Fig. S5G). This rules out ETC expression from explaining the difference in lifespan in 0.5 mM potassium, but does not get to your question directly.

To directly address your question, we performed the same experiment shown in Fig. 7A on mitochondria isolated from *sis2Δ* mutants (Fig. S5I) and found that, as with WT, MMP declines with increasing KCl in the buffer. Thus, even if the protein composition of *sis2Δ* mitochondria is different from WT mitochondria, our fundamental finding that MMP is affected by potassium still holds.

The authors propose that deletion of SIS2 results in increased Ppz1 activity. Is PPZ1 overexpression sufficient to increase MMP and lifespan?

This is a good point and we tried several ways of overexpressing *PPZ1*. We tried overexpressing *PPZ1* by integrating an extra copy with a weak promoter (prADH1 or prCYC1), but it had no effect on MMP or lifespan. We also tried titrating the amount of *PPZ1* with a doxycycline-inducible promoter, but any amount of induction led to shorter-lived strains. As we mention in the text, overexpressing Ppz1p can be toxic to the cell (Makanæ *et al.*, 2013; Casamayor and Ariño, 2022), and has been shown to affect ~20% of the genome (Velázquez *et al.*, 2020). As we show in Fig. 2E, deleting *PPZ1* also leads to low MMP and short lifespans.

It is important to remember that Sis2p acts by inhibiting the phosphatase activity Ppz1p, perhaps for only some substrates, and not by changing the expression of *PPZ1*. Thus our inability to recapitulate the activity of Ppz1p in a *sis2Δ* cell by manipulation of *PPZ1* expression is perhaps not surprising.

Given that sat4 and hal5 effects on lifespan and MMP are additive to sis2, suggests they are not functionally linked. Rather, a common function would suggest an epistatic relationship, which the authors seem to not observe.

Additive effects do not necessarily imply lack of functional linkage. In our model (Fig. S5K), the kinases Sat4p/Hal5p act to increase the activity of Trk1p, while the phosphatase Ppz1p acts to reduce the activity of Trk1p. Sis2p indirectly modulates the activity of Trk1p through its inhibition of Ppz1p. Therefore, *SAT4/HAL5* and *SIS2* are functionally linked by their similar effects on the activity of Trk1p. These are not “classically” defined epistatic effects because their effects on Trk1p are quantitative: deleting *SAT4/HAL5* decreases the activity of Trk1p, while deleting *SIS2* dysinhibits Ppz1p, which leads to even less active Trk1p.

Minor points

Figure S2F: there are multiple datapoints labeled as "WT". What do they represent?

That figure showed the results of several experiments, each with their own WT control, on the same plot. We re-analyzed the data that went into this figure to incorporate the fact that the same strains were measured multiple times over several experiments. In the new figure (Fig. S3F), each genotype is now represented by a single point.

Figure 4A and B need include the sis2trk1 double mutant.

Thank you for the suggestion. We have added this genotype to these figures and we believe this makes our conclusion that *sis2*Δ is a hypomorphic allele of Trk1p even stronger.

Does intracellular potassium change with cellular age? Cell volume increases with age. Does this affect the intracellular potassium concentration?

We compared the internal potassium of aged vs young cells, and found that, while aged cells have more total potassium than young cells, the overall increase in volume means the concentration of potassium in old cells is significantly lower (Fig. S5E). There are many reasons this could be the case. For example, cells might reduce their internal potassium levels in an attempt to rescue their failing mitochondria, but the safest conclusion is that the correlation between potassium levels and MMP does not hold in aged cells (lines 361-364). We note that this does not affect any of the conclusions in the manuscript.

Does intracellular potassium affect vacuolar acidity during aging? The authors test the effects on the vacuole in young but not in aged cells (Figure S4). This is an important control.

This is an interesting question, but it would require substantial methods development to achieve, given that we don't know how the ratiometric vSEP system (vacuole pH reporter) works when proteostasis starts to decline in old age. We have added text to the discussion to point out that changes to vacuolar pH with age might interact with the observations we make in our current study.

Reviewer #2

Specific points:

1. The title claims that potassium ion homeostasis regulates mitochondrial function. First of all, the authors did not show that potassium regulates, i.e. controls, mitochondrial activity. They just show that the potassium concentration in the medium or in the buffer influences respiration. This would presumably also be the case if the authors would have altered the levels of magnesium, sodium or ammonium.

We are sorry that we didn't clearly explain a fundamental conclusion from our work. We indeed show that isolated mitochondria respond to ionic strength, and NOT potassium in particular (Fig. 7C,F and Fig. S9B, lines 415-421, lines 436-437, lines 502-509). Potassium is so critical here because its intracellular physiological range corresponds to the range of ionic strength that the mitochondria responds to. Magnesium, sodium, and ammonium are found at concentrations 1-2 orders of magnitude less within cells (Mulikidjanian *et al.*, 2012; Cueto-Rojas *et al.*, 2016).

2. The green-over-red screen to visualize Cox4-over-Tim50 ratios is interesting as it suggests that mitochondria alter the protein import efficiency on a protein specific level. However, the authors just use this interesting aspect as a proxy for membrane potential which they could have analyzed much easier with a membrane potential-sensitive dye.

We appreciate the reviewer's comments and this led us to explain in greater detail the limitations of an MMP-sensitive dye within the text (lines 79-86). Specifically, we desired to have longitudinal monitoring of MMP throughout a cell's lifespan and avoid the numerous technical and biological complications of dyes in a genome-wide screening platform.

It is not clear whether the lower import of Cox4 is due to a reduced membrane potential or vice versa, the reduction in the import of OXPHOS enzymes such as Cox4 reduces the membrane potential. This initial part of the study is very superficial. A more comprehensive analysis why or how the import of some proteins changes during aging would have made this study much more interesting.

We have taken the reviewer's comments to heart and provide a much deeper explanation and justification for the reporter system (lines 77-106). We also hope it is clearer now that we are only using the leader sequence of COX4 (pre-COX4) to target the mNEON green protein to the mitochondria. This method for MMP-sensitive targeting is well-established. Please see (Veatch *et al.*, 2009; Fehrmann *et al.*, 2013; Vowinckel *et al.*, 2015) which we cite in the manuscript. It is true that loss of membrane potential would be expected to reduce import of proteins essential for the function of the mitochondria, and is interesting to consider. However, our study here is focused on demonstrating the role of potassium homeostasis in regulating MMP – a precursor to protein-specific import into the mitochondria.

3. It is not entirely clear how the green-over-red screen works. The authors refer to Martin Ralser's study who however used different fusion proteins. The changes in signal intensities could be due to changes in protein stability rather than membrane potential-dependent import. It is essential to carefully analyze what this reporter exactly measures.

We now offer a much more detailed explanation and justification of the reporter system we employed here (lines 77-106), as both Reviewers #1 and #2 requested.

4. The authors need to exclude that the decline of the green-over-red signal is caused by a different bleaching behavior of the two proteins. They might simply use dead cells for which the ratio would have to stay constant over time.

We have added two figures to further characterize the reporter system. Fig. S1A,B compares the MMP-dependent reporter to the MMP-sensitive dye TMRM. The signal of the MMP_s reporter declines with age, even though each cell was only imaged once. Thus, the decline is not due to photobleaching. Similarly, in the new Fig. S2 we show that the Tim50-mCherry signal does not change with age/time and thus photobleaching is not affecting this portion of the reporter system either. While we agree that, in principle, photobleaching could contribute to a decline in signal intensity, it is not necessary for our reporter system to function. We also compared the expression of the promoters of each of the reporters and found that they do not change over the 24 hour period of interest (Table S1 and lines 648-651).

5. In the supplemental table, the authors provide the specific values of their fluorescence measurements in a test set of different mutants. The table lacks a description, but from what I can see, the total fluorescence intensities in the mutants are extremely heterogeneous. For example, a deletion of DEG1 shows a 10 fold higher log fold change of the GFP signal than the SIS2 mutant that was further analyzed in the study. Does this mean that the Cox4 fusion protein is accumulating in the absence of DEG1 to about 1,000 fold higher levels than in the SIS2 deletion mutant? How can a reporter be reliable if the expression intensities differ so much? Even ratiometric sensors are only reliable if the expression intensities are comparable.

We agree that the table (now Table S2) needs more description and apologize for the confusion. These data represent the LFC of enrichment of the barcodes, as sorted from the “high GFP ratio” vs “low GFP ratio” channels of our FACS data. We have updated Fig. S3A, simplified Table S2, and added a tab to Table S2 explaining what each of the columns is describing.

6. The authors used glucose as carbon source for their experiments. However, many mitochondrial proteins are glucose-repressed. The screen did not bring up any mutant that lacks a mitochondrial protein but many mutants that affect metabolism. Doesn't this suggest that the screen does not report about the age-dependent import behavior of Cox4 and Tim50, but rather about changes in the metabolism of aging cells which affects mitochondrial function only indirectly?

Again, we apologize that Table S2 was not clear. We have added a column with a brief description of gene function, as well as a column indicating whether or not it has a known mitochondrial function. We hope now it is more clear that two of the mutants included in Table S2 (previously Table S1), *mic19Δ* and *hem25Δ*, are deletions of ORFs of known mitochondrial proteins. In addition, there are four other genes (*PSP2*, *FAR3*, *DEP1*, and *IME4*) with mitochondrial-related functions according to the *Saccharomyces* Genome Database (yeastgenome.org).

Since this was an unbiased screen, we make no claims about whether the mutants we found have a “direct” or “indirect” effect on the age-associated decline in MMP ratio.

7. The authors propose that potassium influences cytochrome oxidase activity and that this is the basis of the age-dependent decline in mitochondrial function. It should be easy to measure cytochrome oxidase activity or levels in young vs old cells in wild type and TRK1 mutants.

Thank you for the suggestion. We were able to show that ionic strength affects cytochrome c oxidase activity in young cells (Fig. 7F). The ability to isolate functional mitochondria from aged yeast cells is technically very challenging, so we are unable to fulfill the second portion of this suggestion.

While we still conclude that potassium influences cytochrome oxidase activity and MMP, we are aware of several other contributing factors (beyond potassium effects) to mitochondrial dysfunction with age (Hughes and Gottschling, 2012; Hughes *et al.*, 2020), and did not mean to imply potassium levels are the sole driver of age-associated loss of MMP.

Reviewer #3

Specific points

1. The authors should investigate whether the cytoplasmic potassium ion levels are high in aged wild-type cells. If that is the case, which factor(s) could be the primary cause of alterations in potassium ion levels?

Thank you for the suggestion - this was also recommended by Reviewer #1. Consequently, we have performed these experiments; please see above in our responses to Reviewer #1.

2. It would be interesting to see if loss of Sis2 can ameliorate short lifespan phenotypes of mutants that are unrelated to potassium ion homeostasis.

We agree that a study on the interactions between *sis2*Δ and other mutants would be interesting, we envision this to be a future research direction.

3. The authors should try to test if the OXPHOS complex IV activity is indeed increased in cells lacking Sis2.

While this is a good idea, we cannot make *in vivo* measurements of complex IV activity. However, we were able to show that the MMP of mitochondria isolated from *sis2*Δ also decreases with increasing potassium concentration in the buffer (Fig. S5I). We also found that complex IV activity is sensitive to the ionic strength of the buffer (Fig. 7F). Thus, since the

internal potassium of *sis2Δ* mutants is lower than WT in regular media (Fig. 4A, Fig. 6A), we believe that the activity of *sis2Δ* complex IV will be higher.

References

- Casamayor, A. and Ariño, J. (2022) “When Phosphatases Go Mad: The Molecular Basis for Toxicity of Yeast Ppz1,” *International Journal of Molecular Sciences*, 23(8), p. 4304. Available at: <https://doi.org/10.3390/ijms23084304>.
- Cueto-Rojas, H.F. *et al.* (2016) “In Vivo Analysis of NH₄⁺ Transport and Central Nitrogen Metabolism in *Saccharomyces cerevisiae* during Aerobic Nitrogen-Limited Growth,” *Applied and Environmental Microbiology*, 82(23), pp. 6831–6845. Available at: <https://doi.org/10.1128/AEM.01547-16>.
- Fehrmann, S. *et al.* (2013) “Aging Yeast Cells Undergo a Sharp Entry into Senescence Unrelated to the Loss of Mitochondrial Membrane Potential,” *Cell Reports*, 5(6), pp. 1589–1599. Available at: <https://doi.org/10.1016/j.celrep.2013.11.013>.
- Gilon, T., Chomsky, O. and Kulka, R.G. (1998) “Degradation signals for ubiquitin system proteolysis in *Saccharomyces cerevisiae*,” *The EMBO Journal*, 17(10), pp. 2759–2766. Available at: <https://doi.org/10.1093/emboj/17.10.2759>.
- Herrera, R. *et al.* (2013) “Subcellular potassium and sodium distribution in *Saccharomyces cerevisiae* wild-type and vacuolar mutants,” *Biochemical Journal*, 454(3), pp. 525–532. Available at: <https://doi.org/10.1042/BJ20130143>.
- Hughes, A.L. *et al.* (2016) “Selective sorting and destruction of mitochondrial membrane proteins in aged yeast,” *eLife*, 5. Available at: <https://doi.org/10.7554/eLife.13943>.
- Hughes, A.L. and Gottschling, D.E. (2012) “An early age increase in vacuolar pH limits mitochondrial function and lifespan in yeast,” *Nature*, 492(7428), pp. 261–265. Available at: <https://doi.org/10.1038/nature11654>.
- Hughes, C.E. *et al.* (2020) “Cysteine Toxicity Drives Age-Related Mitochondrial Decline by Altering Iron Homeostasis,” *Cell*, 180(2), pp. 296–310.e18. Available at: <https://doi.org/10.1016/j.cell.2019.12.035>.
- Makanae, K. *et al.* (2013) “Identification of dosage-sensitive genes in *Saccharomyces cerevisiae* using the genetic tug-of-war method,” *Genome Research*, 23(2), pp. 300–311. Available at: <https://doi.org/10.1101/gr.146662.112>.
- Mulkidjanian, A.Y. *et al.* (2012) “Origin of first cells at terrestrial, anoxic geothermal fields,” *Proceedings of the National Academy of Sciences*, 109(14), pp. E821–E830. Available at: <https://doi.org/10.1073/pnas.1117774109>.
- Papić, D. *et al.* (2013) “The Role of Dj p1 in Import of the Mitochondrial Protein Mim1 Demonstrates Specificity between a Cochaperone and Its Substrate Protein,” *Molecular and Cellular Biology*, 33(20), pp. 4083–4094. Available at: <https://doi.org/10.1128/MCB.00227-13>.

Shlevin, L. *et al.* (2007) "Location-Specific Depletion of a Dual-Localized Protein," *Traffic*, 8(2), pp. 169–176. Available at: <https://doi.org/10.1111/j.1600-0854.2006.00518.x>.

Veatch, J.R. *et al.* (2009) "Mitochondrial Dysfunction Leads to Nuclear Genome Instability via an Iron-Sulfur Cluster Defect," *Cell*, 137(7), pp. 1247–1258. Available at: <https://doi.org/10.1016/j.cell.2009.04.014>.

Velázquez, D. *et al.* (2020) "Yeast Ppz1 protein phosphatase toxicity involves the alteration of multiple cellular targets," *Scientific Reports*, 10(1), p. 15613. Available at: <https://doi.org/10.1038/s41598-020-72391-y>.

Vowinckel, J. *et al.* (2015) "MitoLoc: A method for the simultaneous quantification of mitochondrial network morphology and membrane potential in single cells," *Mitochondrion*, 24, pp. 77–86. Available at: <https://doi.org/10.1016/j.mito.2015.07.001>.

Yenush, L. (2016) "Potassium and Sodium Transport in Yeast," in J. Ramos, H. Sychrová, and M. Kschischo (eds.) *Yeast Membrane Transport*. Cham: Springer International Publishing (Advances in Experimental Medicine and Biology), pp. 187–228. Available at: https://doi.org/10.1007/978-3-319-25304-6_8.

December 3, 2025

RE: JCB Manuscript #202505110R

Dan Gottschling
Calico Life Sciences LLC

Dear Dr. Gottschling:

Thank you for submitting your revised manuscript entitled "Potassium ion homeostasis regulates mitochondrial function". The reviewers all now support publication so we would be happy to publish your paper in JCB pending final revisions necessary to meet our formatting guidelines (see details below).

In your final revision, please be sure to address reviewer #2's and #3's final minor concerns. We would encourage you to suggest an alternative title for your paper.

A. MANUSCRIPT ORGANIZATION AND FORMATTING:

Full guidelines are available on our Instructions for Authors page, <http://jcb.rupress.org/submission-guidelines#revised>.

1) Text limits: Character count for Articles is < 40,000, not including spaces. Count includes abstract, introduction, results, discussion, and acknowledgments. Count does not include title page, figure legends, materials and methods, references, tables, or supplemental legends.

2) Figures limits: Articles may have up to 10 main text figures.

3) Figure formatting: Scale bars must be present on all microscopy images, including inset magnifications. Molecular weight or nucleic acid size markers must be included on all gel electrophoresis. Aspect ratios of images may not be altered.

****4) Statistical analysis:** Error bars on graphic representations of numerical data must be clearly described in the figure legend. The number of independent data points (n) represented in a graph must be indicated in the legend. Statistical methods should be explained in full in the materials and methods. For figures presenting pooled data the statistical measure should be defined in the figure legends. Please also be sure to indicate the statistical tests used in each of your experiments (either in the figure legend itself or in a separate methods section) as well as the parameters of the test (for example, if you ran a t-test, please indicate if it was one- or two-sided, etc.). Also, if you used parametric tests, please indicate if the data distribution was tested for normality (and if so, how). If not, you must state something to the effect that "Data distribution was assumed to be normal but this was not formally tested."

5) Abstract and title: The abstract should be no longer than 160 words and should communicate the significance of the paper for a general audience. The title should be less than 100 characters including spaces. Make the title concise but accessible to a general readership.

Based on reviewer #2's critique, we encourage you to suggest an alternative title for your paper.

****6) Materials and methods:** Should be comprehensive and not simply reference a previous publication for details on how an experiment was performed. Please provide full descriptions in the text for readers who may not have access to referenced manuscripts. For example, in "Screen media, growth, and aging" paragraph of your methods section, you state: "MADs were loaded in triplicate and run for 24 h ("T24") as described (Hendrickson et al. 2018)." You must remove this and write out the full methodology. The same goes for all other places in the methods where you refer to methods "...as described."

7) All antibodies, cell lines, animals, and tools used in the manuscript should be described in full, including accession numbers for materials available in a public repository such as the Resource Identification Portal. Please be sure to provide the sequences for all of your primers/oligos and RNAi constructs in the materials and methods. You must also indicate in the methods the source, species, and catalog numbers (where appropriate) for all of your antibodies. Please also indicate the acquisition and quantification methods for immunoblotting/western blots.

8) Microscope image acquisition: The following information must be provided about the acquisition and processing of images:
a. Make and model of microscope
b. Type, magnification, and numerical aperture of the objective lenses

- c. Temperature
- d. Imaging medium
- e. Fluorochromes
- f. Camera make and model
- g. Acquisition software
- h. Any software used for image processing subsequent to data acquisition. Please include details and types of operations involved (e.g., type of deconvolution, 3D reconstitutions, surface or volume rendering, gamma adjustments, etc.).

10) Supplemental materials: There are strict limits on the allowable amount of supplemental data. Articles may have up to 5 supplemental figures. Please also note that tables, like figures, should be provided as individual, editable files. A summary of all supplemental material should appear at the end of the Materials and methods section.

**12) Conflict of interest statement: JCB requires inclusion of a statement in the acknowledgements regarding competing financial interests. If no competing financial interests exist, please include the following statement: "The authors declare no competing financial interests." If competing interests are declared, please follow your statement of these competing interests with the following statement: "The authors declare no further competing financial interests."

**13) ORCID IDs: ORCID IDs are unique identifiers allowing researchers to create a record of their various scholarly contributions in a single place. Please note that ORCID IDs are now *required* for all authors. At resubmission of your final files, please be sure to provide your ORCID ID and those of all co-authors.

**14) A separate author contribution section following the Acknowledgments. All authors should be mentioned and designated by their full names. We encourage use of the CRediT nomenclature.

Please note that JCB now requires authors to submit Source Data used to generate figures containing gels and Western blots with all revised manuscripts. This Source Data consists of fully uncropped and unprocessed images for each gel/blot displayed in the main and supplemental figures. For assays performed using capillary electrophoresis and/or immunoassay-based detection, authors should instead provide the electropherogram graph(s) for each experiment, plotting fluorescence/chemiluminescence intensity vs. molecular weight/size. Please be sure to provide one Source Data file for each figure gels, blots, and/or capillary electrophoresis assays along with your revised manuscript files. File names for Source Data figures should be alphanumeric without any spaces or special characters (i.e., SourceDataF#, where F# refers to the associated main figure number or SourceDataFS# for those associated with Supplementary figures). For traditional gels and blots, the lanes of the gels/blots should be labeled as they are in the associated figure, the place where cropping was applied should be marked (with a box), and molecular weight/size standards should be labeled wherever possible. For capillary electrophoresis assays, each trace in the graph should be color-coded and labeled to indicate which protein, gene, or sample is being measured (please try to avoid red/green combinations to accommodate our color-blind readers).

Journal of Cell Biology now requires a data availability statement for all research article submissions. These statements will be published in the article directly above the Acknowledgments. The statement should address all data underlying the research presented in the manuscript. Please visit the JCB instructions for authors for guidelines and examples of statements at (<https://rupress.org/jcb/pages/editorial-policies#data-availability-statement>).

B. FINAL FILES:

Thank you for your attention to these final processing requirements. Please revise and format the manuscript and upload materials within 7 days. If you need an extension for whatever reason, please let us know and we can work with you to determine a suitable revision period.

Thank you for this interesting contribution, we look forward to publishing your paper in Journal of Cell Biology.

Sincerely,

Thomas Langer
Monitoring Editor
Journal of Cell Biology

Gabriele Stephan
Scientific Editor
Journal of Cell Biology

Reviewer #1:

The authors addressed all of my comments and critical points.

Reviewer #2:

The authors addressed several of the points raised on the original submission. I am still not convinced that the potassium ion homeostasis REGULATES mitochondrial function as claimed in the title, nor is the green-over-red assay properly characterized on a mechanistic level. However, as I doubt that a further round of revision would improve the study considerably, I support the publication of this study in its present form. The data might be useful for readers studying cellular aging in yeast.

Reviewer #3:

In this revised manuscript, Waite et al. et al. provided additional data and descriptions to clarify most of the points suggested by the referees. Although it is still not fully understood how potassium homeostasis regulates mitochondrial membrane potential (MPP) and whether a decline in MPP is a cause or consequence of aging, this comprehensive study will promote deep investigations towards molecular understanding of MPP-related aging in the future.